# Core and rod structures of a thermophilic cyanobacterial light-harvesting phycobilisome

Keisuke Kawakami[1,8 ✉], Tasuku Hamaguchi [1,8], Yuu Hirose[2], Daisuke Kosumi [3], Makoto Miyata [4], Nobuo Kamiya[5] & Koji Yonekura [1,6,7 ✉]

Cyanobacteria, glaucophytes, and rhodophytes utilize giant, light-harvesting phycobilisomes (PBSs) for capturing solar energy and conveying it to photosynthetic reaction centers. PBSs are compositionally and structurally diverse, and exceedingly complex, all of which pose a challenge for a comprehensive understanding of their function. To date, three detailed architectures of PBSs by cryo-electron microscopy (cryo-EM) have been described: a hemiellipsoidal type, a block-type from rhodophytes, and a cyanobacterial hemidiscoidal-type. Here, we report cryo-EM structures of a pentacylindrical allophycocyanin core and phycocyanin-containing rod of a thermophilic cyanobacterial hemidiscoidal PBS. The structures define the spatial arrangement of protein subunits and chromophores, crucial for deciphering the energy transfer mechanism. They reveal how the pentacylindrical core is formed, identify key interactions between linker proteins and the bilin chromophores, and indicate pathways for unidirectional energy transfer.

[1] Biostructural Mechanism Laboratory, RIKEN SPring-8 Center, 1-1-1, Sayo, Hyogo 679-5148, Japan. [2] Electronics-Inspired Interdisciplinary Research Institute, Toyohashi University of Technology, 1-1 Tempaku, Toyohashi, Aichi 441-8580, Japan. [3] Institute of Industrial Nanomaterials, Kumamoto University, Kumamoto 860-8555, Japan. [4] Graduate School of Science, Osaka Metropolitan University, Osaka 558-8585, Japan. [5] The OCU Advanced Research Institute for Natural Science & Technology (OCARINA), Osaka Metropolitan University, Osaka 558-8585, Japan. [6] Advanced Electron Microscope Development Unit, RIKEN-JEOL Collaboration Center, RIKEN Baton Zone Program, Hyogo 679-5148, Japan. [7] Institute of Multidisciplinary Research for Advanced Materials, Tohoku University, Miyagi 980-8577, Japan. [8] These authors contributed equally: Keisuke Kawakami, Tasuku Hamaguchi. ✉email: kawakami.k@spring8.or.jp; yone@spring8.or.jp

Cyanobacteria, glaucophytes, and rhodophytes utilize a large water-soluble, light-harvesting complex called phycobilisome (PBS) for solar energy absorption and energy transfer to photosynthetic membrane proteins (photosystem I and photosystem II; PSI and PSII)[1]. PBSs absorb mainly visible light at 490–650 nm, which PSI and PSII have difficulty absorbing (as an exception, cyanobacteria containing chlorophyll *f* harbor PBSs that absorb near-infrared light[2,3]) (Supplementary Fig. 1). PBS is composed of phycobiliproteins (PBPs) such as phycoerythrin, phycoerythrocyanin, phycocyanin (PC), and allophycocyanin (APC). Assembly units of the PBPs are α- and β-subunits that have globin folds and harbor several linear tetrapyrrole chromophores such as phycoerythrobilin, phycourobilin, phycoviolobilin, and phycocyanobilin (PCB)[4]. Oligomers of these α- and β-subunits interact with non-chromophorylated linker proteins to form PC rods in the whole PBS complex. Five types of PBS structural morphology have been reported: hemidiscoidal[5–7], hemiellipsoidal[8], block-type[9], rod-type[10–13], and bundle-type[14]. Hemidiscoidal PBSs have bicylindrical[15,16], tricylindrical[5], or pentacylindrical cores[3,17]. The composition of PBPs with respect to the number of rods, the overall structure of the PBSs, and their association with PSI and PSII are regulated by the availability of light and nutrients[11,18]. Recently, the three-dimensional (3D) structures of three types of PBSs were analyzed by cryo-electron microscopy (cryo-EM)[19–22], and details of their tricylindrical and pentacylindrical cores and energy transfer pathways were revealed. Because PBSs are compositionally and structurally diverse and exceedingly complex, structural and functional analyses of PBSs in different species are essential for the comprehensive understanding of their working mechanisms. Moreover, these structures provide a basis for understanding several cyanobacterial PBSs and can be utilized promoting the efficient use of solar energy[23–25].

Here, we report the structures of the pentacylindrical APC core and PC rod from the thermophilic cyanobacterium *Thermosynechococcus vulcanus* at the resolutions of 3.7 Å and 4.2 Å, respectively. Although their overall structures were extremely similar to the PBS structure of *Anabaena* sp. PCC 7120 (hereafter referred to as *Nostoc* 7120) reported by ref. [22], we observed the differences in their internal structures. The structures revealed differences in the hemidiscoidal PBS of different species and provide a possible mechanism for unidirectional excitation energy transfer.

## Results and discussion

**Subunit composition of PBS from *Thermosynechococcus vulcanus*.** The PBS of *T. vulcanus* NIES-2134 (hereafter referred to as *T. vulcanus*) has a hemidiscoidal structure with a pentacylindrical APC core and six PC rods, with a total molecular weight of ~6 MDa[22,26] (Supplementary Fig. 2). The core and rods comprise α- (CpcA, ApcA, ApcD, and ApcE) and β- (CpcB, ApcB, and ApcF) subunits, and their basic constituent unit is an αβ monomer[27]. PCBs are covalently bound to conserved cysteine residues in the α and β subunits via thioether linkages[28]. Linker proteins in the PBS are believed to contribute not only to the structural stability of the PBS but also adjustment of the energy level of the chromophores[29]. Linker proteins of *T. vulcanus* are classified as rod linker ($L_R$; CpcC); rod-terminal linker ($L_{RT}$; CpcD); rod-core linkers ($L_{RC}$; CpcG1, CpcG2, and CpcG4); core-membrane linkers ($L_{CM}$; ApcE), which bind to the APC trimers of the PBS core cylinders; or core linkers ($L_C$; ApcC), which bind to the PBS core. When the PBS absorbs light, the excitation energy is rapidly transferred (on the order of picoseconds) to chromophores in subunits at the membrane surface called terminal emitters (ApcE [$L_{CM}$] and ApcD) and then eventually to PSII and

PSI[30]. Several X-ray crystal structures of PCs and APCs from *T. vulcanus* have been reported, with these having provided starting points for discussions concerning possible overall structural configurations and mechanisms of internal energy transfer in PBSs[31–37].

**Purification, structural refinement, and overall structure of PBS core.** The pentacylindrical APC core (PBS core) and PC rod were obtained from preparations of PBSs, followed by gradient fixation (GraFix) treatment[26,38] and low-concentration phosphate buffer treatment, respectively. For cryo-EM imaging, it is preferable to remove highly concentrated stabilizers (in this case, the stabilizer was a phosphate buffer); however, this promotes the dissociation of PC rods from the PBS core both in red algae and cyanobacteria including *T. vulcanus*. Although PC rods were bound to the PBS core in the prepared PBS, most of the PC rods were dissociated during GraFix treatment (Supplementary Fig. 2). Two-dimensional (2D) averages with negative-staining EM showed that the cylinder at the bottom of the prepared *T. vulcanus* PBS comprises three APC trimers. As reported by ref. [22], the PC rods in PBSs from cyanobacteria are considerably less interactive than those in red algae[19], and cross-linking treatment with glutaraldehyde to prevent the dissociation of the PC rods helped to maintain the structure of the PBS core. However, some of the PC rods dissociated from the PBS core. The reason for the weaker interaction of PC rods in cyanobacterial PBS as compared with red algal PBS is likely due to the differences in PBS morphology. In red algae PBSs (block and hemiellipsoidal types), there are many densely packed phycoerythrin (PE) rods, suggesting the possibility of numerous interactions between the rods. However, the PC rods of cyanobacterial PBS (hemidiscoidal-type) are arranged in a fan shape, indicating that there are less interactions between the PC rods than for the PE rods of the red algae PBSs. The difference in morphology suggests that cyanobacterial PBS is more prone to the dissociation of PC rods than red algae PBS under low-concentration phosphate buffer conditions.

Cryo-EM maps of the PBS core and PC rod were reconstructed to 3.7 Å and 4.2 Å resolutions, respectively, based on the Gold Standard Fourier shell correlation [FSC] criteria of 0.143 between two half maps (Supplementary Table 1). The structural models were refined against the maps, which validated the resolutions; an FSC value of 0.5 between the model and the map: 3.8 Å (PBS core) and 4.2 Å (PC rod); Q-score[39] based estimates: 3.6 Å ($Q = 0.49$; PBS core) and 4.0 Å ($Q = 0.40$; PC rod) (Supplementary Figs. 3 and 4 and Supplementary Tables 1–3).

The analyzed PBS core of *T. vulcanus* shows a hemidiscoidal structure with C2 symmetry, composed of three cylinders (A, A', and B), two cylinders (C and C') with some PC rods interacting with the PBS core (Fig. 1). The overall structure of the PBS core was very similar to that of *Nostoc* 7120[7,22]. The strict twofold symmetry was not held in the outer part of the PC rods (Rb, Rb', Rt, Rt', Rs1, Rs1', Rs2, and Rs2')[7], as many of these parts likely dissociated from the core during sample preparation (see above). The cryo-EM map resolves portions of the rods, but local resolution of the corresponding regions ranges from 7 Å to 20 Å (Supplementary Fig. 3). Thus, we did not build models of the rods (Fig. 1c). The PBS core is a supercomplex with the dimensions $110 \times 210 \times 300$ Å and is composed of 38 ApcAs, 40 ApcBs, 6 ApcCs, 2 ApcDs, 2 ApcEs ($L_{CM}$s), 2 ApcFs, and 84 PCBs (Supplementary Fig. 5, Supplementary Table 4). In the other cyanobacterial PBS core[22], the A and B cylinders each consist of four APC trimers, whereas the PBS core of *T. vulcanus* comprises three APC trimers in the A cylinder and four APC trimers in the B cylinder. ApcD has been identified in the fourth APC trimer of the A cylinder of the other cyanobacterial PBSs, and this trimer

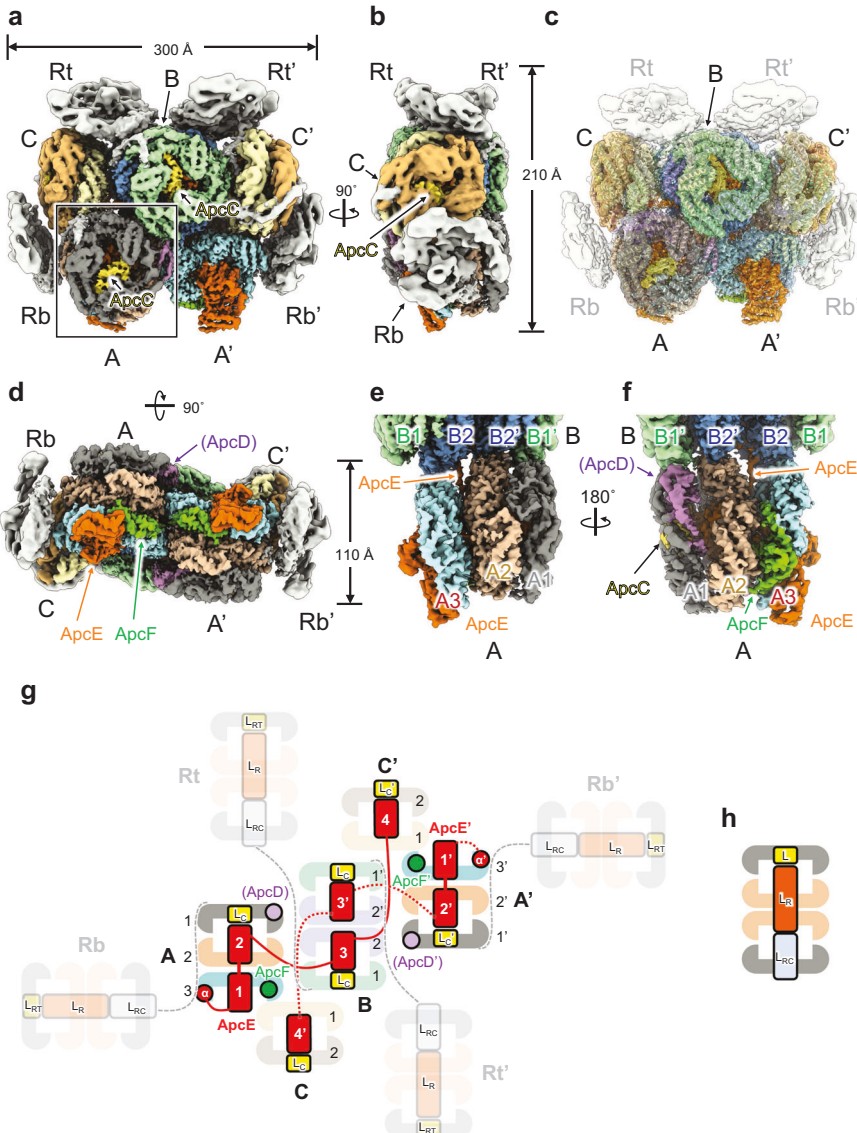

**Fig. 1 Overall structure of the PBS core from *T. vulcanus*.** Cryo-EM map of the PBS core from the front (**a**, **c**), side (**b**, **e**, **f**), and bottom (**d**) views. **c** It shows the superposition of the cryo-EM map and the refined PBS core model. **e** Here it shows the structure inside the rectangle in **a**, enlarged and rotated 90°. **f** This shows **e** rotated 180°. **g** Schematic model of the pentacylindrical APC core (including A (A′), B, and C (C′) cylinders) and PC rods (Rb, Rb′, Rt, and Rt′). ApcE ($L_{CM}$) containing α, Reps 1–4 and Arms 1–3 is shown in red. PC rods that did not build the model are shown in translucent. The PC rod models were drawn referencing the cryo-EM map in this study and the PBS structure from *Nostoc* 7120[7]. **h** Schematic model of the PC rod. CpcC ($L_R$), CpcD ($L_{RT}$), and CpcG ($L_{RC}$) interact within the two PC hexamers. The identification of ApcD is tentative. ApcD is shown in parentheses (ApcD).

plays an important role in energy transfer from PBS to photosynthetic membrane proteins[22]. It is considered that if the arrangement of ApcD is same in the PBSs of the cyanobacteria *Synechococcus* sp. PCC 7002 (hereafter referred to as *Synechococcus* 7002), *Nostoc* 7120, and *T. vulcanus*, then *T. vulcanus* PBS should have a fourth APC trimer in the A cylinder. However, the prepared PBS of *T. vulcanus* contains ApcD, and 2D averages with negative-staining EM confirmed that the A cylinder comprised of three APC trimers (Supplementary Fig. 2a, c, d). This suggests that the arrangement of ApcD in *T. vulcanus* PBS differ from that of PBS in other species. At present, we do not clearly understand why *T. vulcanus* has three APC trimers in the A cylinder rather than four.

The A cylinder is composed of the α-subunits of phycobiliproteins (ApcA, ApcD, and ApcE [$L_{CM}$]), the β-subunits of phycobiliproteins (ApcB and ApcF), and ApcC (Lc). The A (A′) cylinder consists of three APC trimers (A1, A2, and A3) with

subunits of ApcD/ApcB and two ApcA/ApcB (A1); three ApcA/ApcB (A2); and ApcE/ApcB, ApcA/ApcF, and ApcA/ApcB (A3), respectively (Fig. 2a). Electrophoretic analysis shows that ApcD is present in prepared PBSs (Supplementary Fig. 2c)[26], but ApcD could not be identified in the cryo-EM map due to the limited resolution. We then tentatively assigned ApcD to a subunit that could not be identified as ApcA in this study (however, we identified ApcA rather than ApcD in the structural model [PDB code: 7VEA]) (Supplementary Fig. 6). The B cylinder consists of four APC trimers consisting of ApcA and ApcB (B1, B2, B1′, and B2′). In addition, $L_C$ interacts with one side of the A and C cylinders, while it interacts with both sides of the B cylinder. The C cylinder consists of two APC trimers, ApcA and ApcB, which correspond to a half B cylinder.

ApcE ($L_{CM}$) is a terminal emitter that is located at the membrane surface in the A cylinder and transfers energy to PSII. ApcE ($L_{CM}$) consists of $α^{LCM}$, and one of its domains

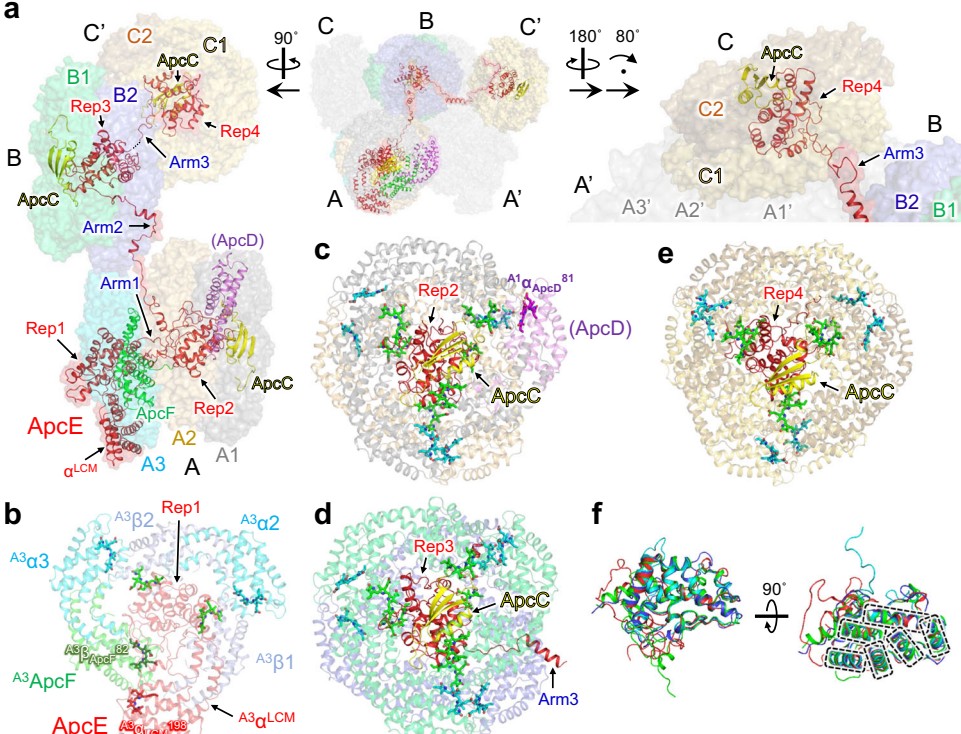

**Fig. 2 Structures of A, B, and C cylinders, including terminal emitters and linker proteins. a** Arrangement of the terminal emitters ($\alpha^{LCM}$ of ApcE [$L_{CM}$] and ApcD), ApcF, and linker proteins (ApcE [$L_{CM}$] and ApcCs). **b** Structures of APC trimer (A3) and Rep1 of ApcE ($L_{CM}$) in A cylinder. **c** Structures of APC trimers A1 and A2, ApcC, and Rep2 of ApcE ($L_{CM}$) in A cylinder. **d** Structures of APC trimers B1 and B2, ApcC, Rep3, and Arm3 of ApcE ($L_{CM}$) in B cylinder. **e** Structure of APC trimers C1 and C2, ApcC, and Rep4 of ApcE ($L_{CM}$) in C′ cylinder. **f** Superposition of four Rep structures (Reps1–4). Dashed lines indicate the α-helices in the Rep regions. Rep1, green; Rep2, red; Rep3, blue; Rep4, cyan.

exhibits a similar structure to ApcA, four repeated motifs named "repeats" (Rep1–Rep4), and "arms" (Arm1–3) that connect the motifs. Rep1 interacts with ApcF, two ApcAs (α2 and α3), and one ApcB (β1) in A1 (Fig. 2b). Although the ApcEs ($L_{CM}$s) of the two red algae PBSs and *Synechococcus* 7002 PBS do not contain Rep4, the root-mean-square deviations (RMSDs) of Rep1–Rep4 in the ApcEs ($L_{CM}$s) of each species are small, and the sequence identity of Rep1–Rep4 from cyanobacteria tends to be high (Supplementary Table 5). This suggests that the structure and function of each Rep is similar, despite the difference in species.

In the recently reported structures of red algae, ApcC interacts with Rep1 of ApcE ($L_{CM}$), whereas ApcC and (ApcD) might interact with Rep2 of the PBS core of *T. vulcanus*, suggesting that the interaction between subunits in the core also differs between species. Rep3 of ApcE ($L_{CM}$), together with ApcC, contribute to the maintenance of the structure and function of the two APC trimers in the B cylinder. Arm3 extends to Rep4, which serves as the linker domain of the C cylinder. Rep4 interacts with ApcC in the same way as Rep2 and Rep3, which means that each cylinder in the PBS core contains one ApcC (Fig. 2c–e). An interesting feature of Reps1–4 in ApcE ($L_{CM}$) is that the structures of the α-helix regions are similar, whereas the structures of the loop regions differ (Fig. 2f). Sequence identity among Reps is not high (Supplementary Fig. 7), suggesting not only that Reps1–4 maintain the structure of each cylinder but also that distinct amino acid residues in each Rep around PCB could contribute to the efficient transfer of absorbed light energy to the terminal emitters. Arm3, Rep4 of ApcE ($L_{CM}$), and the C cylinder are not present in the PBS core of red algae[19,20] or *Synechocystis*[6], whereas these components are present in about half of cyanobacteria if the ApcE ($L_{CM}$) domains are mapped to the

phylogenetic tree of major cyanobacterial species (Supplementary Figs. 8 and 9). Thus, common to many species of algae including cyanobacteria, the C cylinder, Rep4, and Arm3, which maintain the C cylinder itself, must play a key role in light-harvesting. ApcE ($L_{CM}$) is one of the most important linker proteins for energy transfer to PSII[1]. The structure of *T. vulcanus* PBS described in this study provides a starting point towards understanding component architecture and organizational networks for general working mechanisms of PBSs in the broad algae group.

The lack of the "fourth APC trimer" in the A cylinder of *T. vulcanus* PBS may be attributed to the fact that ApcC interacting in the APC trimer is unable to interact with Rep1 in ApcE. There are two main conformation types (Type I and II) in ApcC of the PBS present in *Nostoc* 7120 and *Synechococcus* 7002. ApcC that interacts with Reps2–4 is type I, whereas the ApcC that interacts with Rep1 is Type II (Supplementary Fig. 10). The structures of ApcC (Types I and II) of *Nostoc* 7120 and *Synechococcus* 7002 are similar (RMSD: 0.78 [Type I] and 0.74 [Type II]). In contrast, in *T. vulcanus* PBS, ApcC interacting with Reps3–4 is Type I, but ApcC interacting with Rep2 is neither Type I nor II (that is, Type III). Comparison of Types II and III revealed different arrangement of the loop regions in ApcC (residues 9–26, Supplementary Fig. 10c), and slightly larger RMSD between Types I and III (1.7). Although the amino acid sequences of Reps1–4 and ApcC are similar in the three cyanobacteria (Supplementary Fig. 10d), the environment around Rep2 in *T. vulcanus* PBS may alter the interaction of ApcC compared with that in the other cyanobacteria. This suggests that the environment surrounding Rep1 in *T. vulcanus* PBS does not allow ApcC interaction, and hence the "fourth APC trimer" is unable to interact with Rep1.

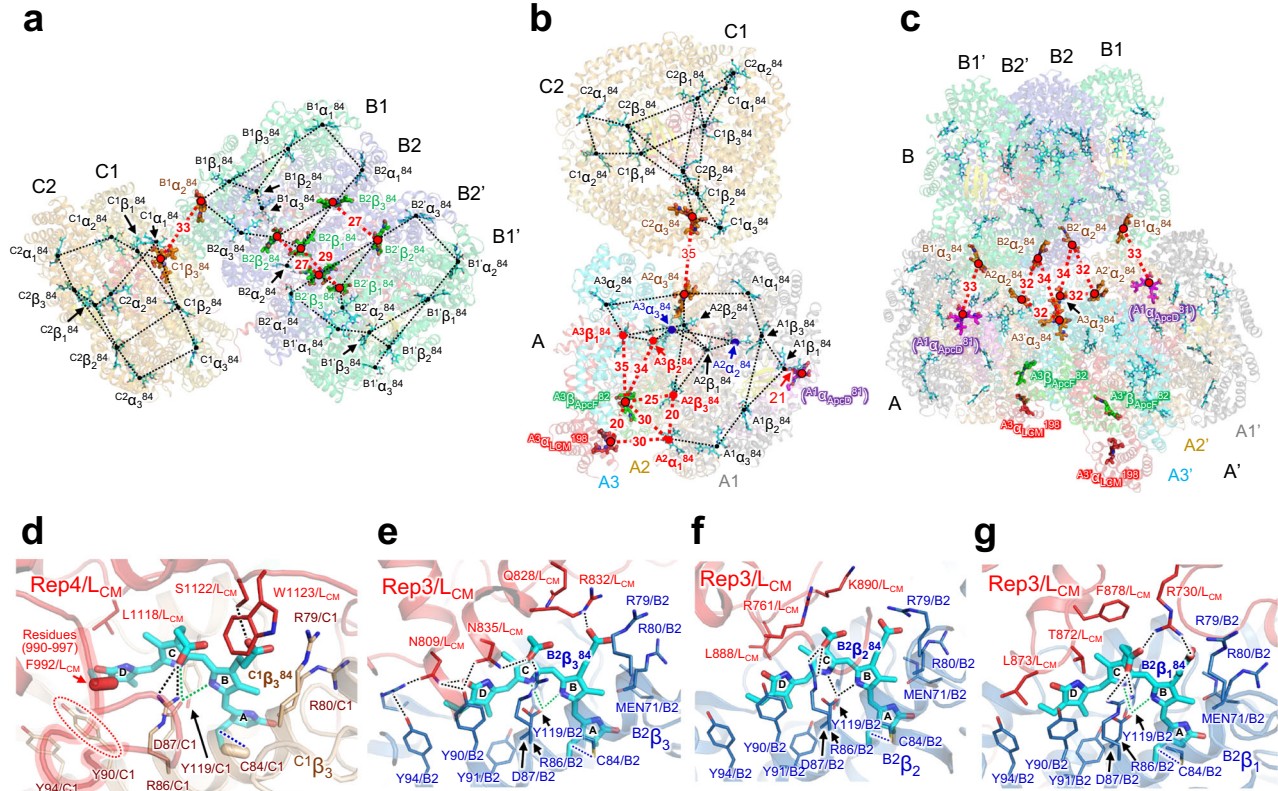

**Fig. 3 Arrangement of chromophores is the key to the energy transfer pathway in PBS core. a** Possible energy transfer pathways in and between B and C (C') cylinders. **b** Possible energy transfer pathways in and between A and C cylinders. **c** Possible energy transfer pathways between A (A') and B cylinders. **d–g** The chromophores and the surrounding amino acid residues of ApcE (L_CM). The numbers near the dotted lines indicate the distances (Å) between the PCB pairs. In **e–g**, dashed lines (black and green) indicate hydrogen bonds, and dotted lines (blue) indicate covalent bonds between PCB and cysteine residue. In the amino acid residues (Phe992/ApcE and Cys84/C1) in **d**, the directions of atoms from $C_\alpha$ to $C_\beta$ in the residues are indicated by the capsule-shaped objects. The identification of ApcD is tentative, and the chromophore in ApcD ($^{A1}\alpha_{ApcD}^{81}$) is indicated in parentheses.

**Arrangement of chromophores in the PBS core and energy transfer pathway.** The excitation energy is transferred to the terminal emitters (ApcE [L_CM] and ApcD) via chromophores in the B and C (C') cylinders, and the energy is eventually transferred to PSII and PSI. *T. vulcanus* has PCB as its only chromophore, and it is the protein environment around each chromophore that must be intimately involved in the transfer of the excited energy to the terminal emitters. Indeed, amino acid residues with polar/charged groups affect the absorption energy of pigments[40].

Figure 3 shows each cylinder that constitutes the PBS core, the chromophores in the core, and the distances between chromophores. In general, the excitation energy is transferred between close pairs of chromophores, i.e., the donor (D) and acceptor (A) (Förster Resonance Energy Transfer; FRET)[41,42]. However, the energy transfer between chromophores in the PBS cannot be explained by the FRET mechanism alone, because photoexcited coherence signals indicating exitonic coupling have been observed in both PC and APC[43,44]. In addition, recent experiments revealed a new characteristic, in which excitations are coherently shared between donor and acceptor molecules during energy transfer[45]. Moreover, previous studies suggest that excitonic coupling of some form may occur, even at distance ranging from 20 Å to 30 Å, enabling the PBS to transfer energy at high efficiencies[29,46]. MacColl[46] reported an excitonic coupling between the two chromophores of the APC trimer located in proximity across the monomer–monomer interface, and this strong interaction induces exciton to a redshift in the absorption spectrum. Although evidence supports both mechanisms, the

ultrafast fluorescence resulting from a number of approaches now indicates that the excitonic coupling mechanism is more plausible; however, although the structural model obtained in this study allows an interpretation of the arrangement and orientation of chromophores in the PBS and their surrounding interactions (especially with linker proteins), the structural model alone does not allow for a comprehensive interpretation of the excitonic coupling between neighboring chromophores. The linker protein not only stabilizes the PBS structure but also alters the absorption/emission properties of the chromophore and may even create the necessary environment for exciton binding between chromophores[29]. In this study, we propose the energy transfer pathway based on the distance between chromophores and their orientation on the chromophores that interact with the linker proteins.

There are four main factors associated with the excitation energy transfer rate by the FRET mechanism: (i) the distance between D and A; (ii) the orientation factor, $\kappa^2$, which is a factor that describes the relative orientation of the transition dipoles of D and A in space (Supplementary Fig. 11); (iii) the overlap integral between the fluorescence and absorption spectra of D and A, respectively; and (iv) quantum yield of D in the absence of A. For a freely rotating pigment D and A, $\kappa^2$ has a mean value of 2/3, and the values of $\kappa^2$ representing the significant chromophores are approximately 1–3 (Supplementary Table 6, Supplementary Figs. 11 and 12). This indicates that energy transfer is likely to occur between their chromophores in the PBS core. The efficiency is inversely proportional to the sixth power of the distance between D and A through dipole–dipole coupling; thus, the

shorter the distance between chromophores, the greater the rate and efficiency of energy transfer between them. Therefore, energy transfer efficiency between cylinders is also strongly dependent on the distance between close chromophores.

The chromophores in the PBS core are numbered according to the nomenclature of those in red alga[19,20], and their names are defined as follows: cylinder name (A, B, or C), subunit type (α or β), and the number of cysteine residue (84) to which the chromophore is bound (Supplementary Fig. 13). Again, we reiterate that the previously described red algal PBS structures do not contain C cylinders[19,20]. The chromophores involved in the energy transfer between or within each cylinder are shown in Fig. 3a–c. The energy transfer from the C (C') cylinder to the B and A (A') cylinders occurs via $^{C1}\alpha_1{}^{84}$–$^{B1}\alpha_2{}^{84}$ (33 Å) and $^{C2}\alpha_3{}^{84}$–$^{A2}\alpha_3{}^{84}$ (35 Å), respectively (Fig. 3a, b). In the present structure, the D ring of PCB in many ApcBs forms π–π interactions with tyrosine residues (Y90), but the D ring of $^{C1}\beta_3{}^{84}$ likely interacts with Phe992 within the loop region (residues 990–997) of Rep4 in the ApcE (Fig. 3d). In addition, $^{C1}\beta_3{}^{84}$ interacts with S1122/ApcE. These interactions probably contribute to energy transfer from the C (C') cylinder to the B cylinder. In the cyanobacterial PBS with bound PC rods (Rt, Rb, Rs1, and Rs2)[7,22,26], the energy received by the PC rods is believed to be transferred to the B and A (A') cylinders via the C (C') cylinder, indicating that the C (C') cylinder acts as an intermediary for energy transfer from the PC rods to the terminal emitters.

The B cylinder consists of four APC trimers, with two APC trimers (B2 and B2') interacting with each other. This indicates that energy transfer in the B cylinder is mediated by $^{B2}\beta_2{}^{84}$–$^{B2'}\beta_1{}^{84}$ (27 Å), $^{B2}\beta_1{}^{84}$–$^{B2'}\beta_1{}^{84}$ (29 Å), and $^{B2}\beta_3{}^{84}$–$^{B2'}\beta{}^{84}$ (27 Å) (Fig. 3a). Energy transfer from the B cylinder to the A (A') cylinder seems to involve $^{B1'}\alpha_3{}^{84}$–$^{A1}\alpha_{ApcD}{}^{81}$ (33 Å) and $^{B2}\alpha_2{}^{84}$–$^{A3'}\alpha_3{}^{84}$ (34 Å) (Fig. 3c). In addition, there is an interaction between the A and A' cylinders, suggesting energy transfer occurs through $^{A2}\alpha_2{}^{84}$–$^{A3'}\alpha_3{}^{84}$ (32 Å). These chromophores interact with Rep3 and Rep4 of ApcE, and the amino acid residue compositions of ApcE differ around each of the four chromophores ($^{C1}\beta_3{}^{84}$, $^{B2}\beta_3{}^{84}$, $^{B2}\beta_2{}^{84}$, and $^{B2}\beta_1{}^{84}$) (Fig. 3d–g). Three chromophores ($^{B2}\beta_3{}^{84}$, $^{B2}\beta_2{}^{84}$, and $^{B2}\beta_1{}^{84}$) interact with amino acid residues in Rep3/ApcE, significantly altering the peripheral structure of each chromophore. $^{B2}\beta_3{}^{84}$ forms hydrogen bonds with asparagine residues (N809/ApcE and N835/ApcE), and basic amino acids (R761/ApcE and K890/ApcE) are present near $^{B2}\beta_2{}^{84}$. $^{B2}\beta_1{}^{84}$ interacts with a basic residue (R730/ApcE) and is surrounded by hydrophobic amino acids (L873/ApcE, T872/ApcE, and F878/ApcE). This difference in the protein environment around each chromophore may define their individual energy level.

**Excitation energy transfer to terminal emitters**. Three PCBs in the A cylinder ($^{A2}\alpha_3{}^{84}$, $^{A2}\alpha_2{}^{84}$, and $^{A3}\alpha_3{}^{84}$) are believed to be the key chromophores that receive excitation energy from the B and C cylinders based on their spatial arrangement (Fig. 3b), which then ultimately pass it to the chromophore ($^{A3}\alpha_{LCM}{}^{198}$) in the terminal emitters. In this case, the excitation energy is most likely transferred via the four chromophores ($^{A2}\alpha_1{}^{84}$, $^{A2}\beta_3{}^{84}$, $^{A3}\beta_1{}^{84}$, and $^{A3}\beta_2{}^{84}$) close to these two chromophores. Three of the four chromophores interact with Rep1 or Rep2, in a characteristic structure (Fig. 4a–c). The κ² value and the distance between chromophores also suggest that the energy transfer efficiency of $^{A3}\alpha_{LCM}{}^{198}$–$^{A3}\beta_{ApcF}{}^{82}$ is highest among all the identified pairs (Supplementary Table 6).

$^{A3}\beta_1{}^{84}$ interacts with Y306/ApcE and is surrounded by aromatic amino acids (Y428/ApcE and F432/ApcE). $^{A3}\beta_2{}^{84}$ interacts with S348/ApcE and R366/ApcE, and F373/ApcE is close to the D ring of $^{A3}\beta_2{}^{84}$. In addition, $^{A2}\beta_3{}^{84}$ interacts with R630/ApcE and is surrounded by three aromatic amino acids (Y455/ApcE, Y610/ApcE, and F637/ApcE), with F637/ApcE close to the D ring of $^{A2}\beta_3{}^{84}$. It was noted that, in the structure of red algal PBSs, aromatic amino acids are abundant in the vicinity of the chromophores and it was proposed that they regulate the energy state of each chromophore to achieve unidirectional energy transfer[19,20]. The aromatic amino acids in the linker proteins of T. vulcanus PBS may function similarly, suggesting that the protein environment around the three chromophores is optimized for efficient transfer of excitation energy to the two chromophores.

The chromophore $^{A3}\beta_{ApcF}{}^{82}$ in ApcF binds C82/ApcF as well as C416/ApcE and interacts with four amino acid residues (Y265/ApcE, R84/ApcF, R77/ApcF, and R78/ApcF). This interaction may aid tuning the energy level of $^{A3}\beta_{ApcF}{}^{82}$ to form the energy transfer pathway from $^{A3}\beta_{ApcF}{}^{82}$ to $^{A3}\alpha_{LCM}{}^{198}$. The environment around $^{A3}\beta_{ApcF}{}^{82}$ in ApcF and $^{A3}\alpha_{LCM}{}^{198}$ in ApcE may be one of the critical elements involved in transferring the unidirectional excitation energy to PSI and PSII. Soulier and Bryant reported that the interactions between PCBs bound to α and β subunits of adjacent APC monomer in APC trimer are responsible for the red-shift in the chromophore absorption[47]. In the T. vulcanus PBS, the distance between $^{A3}\beta_{ApcF}{}^{82}$ and $^{A3}\alpha_{LCM}{}^{198}$ is 20 Å, suggesting that the two chromophores could form an excitonic coupling that eventually transfers excitation energy to PSI and PSII. ApcD is a terminal emitter that transfers energy to PSI[48–50], and the findings of the present study suggest that $^{A1}\alpha_{ApcD}{}^{81}$ forms an excitonic coupling with the neighboring $^{A1}\beta_1{}^{84}$ and transfers excitation energy to PSI.

We were not able to identify the side chains of three amino acid residues (W166/ApcE, D163/ApcE, and K159/ApcE), as the cryo-EM map around $^{A3}\alpha_{LCM}{}^{198}$ was somewhat disordered (Fig. 4e and Supplementary Fig. 3e). Nevertheless, the present structure suggests that $^{A3}\alpha_{LCM}{}^{198}$ in ApcE could form a π–π interaction with W166/AcpE, as the D-ring of PCB and tyrosine residues in ApcA commonly form a π–π interaction. This Tryptophan (Trp) residue is conserved in the ApcE of all the species listed in Supplementary Fig. 9. Tang et al.[51] analyzed the crystal structure (PDB code: 4XXI) of the α region in the ApcE from Nostoc 7120, reporting that the interaction of this Trp with the chromophore is a factor in the red shift in the chromophore adsorption. Moreover, 4XXI is a homodimer, and Trp in one monomer forms a π–π interaction with the D ring of the chromophore, whereas Trp in another monomer is positioned perpendicular to the D ring of the chromophore forming a cation–π interaction with nearby Arg160 (Fig. 5).

To investigate whether π–π interaction is also formed in the PBS structures solved by cryo-EM[19,20,22], we evaluated the α region in the ApcE of the PBS structures and found that only P. purpureum PBS exhibited clear π–π interactions between the D ring of the chromophore and Trp154/ApcE, whereas G. pacifica PBS was modeled to form a π–cation interaction between the D ring of the chromophores and Arg152/ApcE. This is due to the quality of the obtained cryo-EM maps, as the π–π interactions cannot be clearly indicated in the PBS structures, except for that of P. purpureum PBS. However, considering that the π–π interactions contribute to the red shift of the chromophore absorption[51], it is highly likely that the D ring of the chromophore and Trp forms π–π interactions in all PBS structures, indicating that this feature is likely important for energy transfer from $^{A3}\alpha_{LCM}{}^{198}$ to PSII. Further details with regard to $^{A3}\alpha_{LCM}{}^{198}$ and key interactions with surrounding

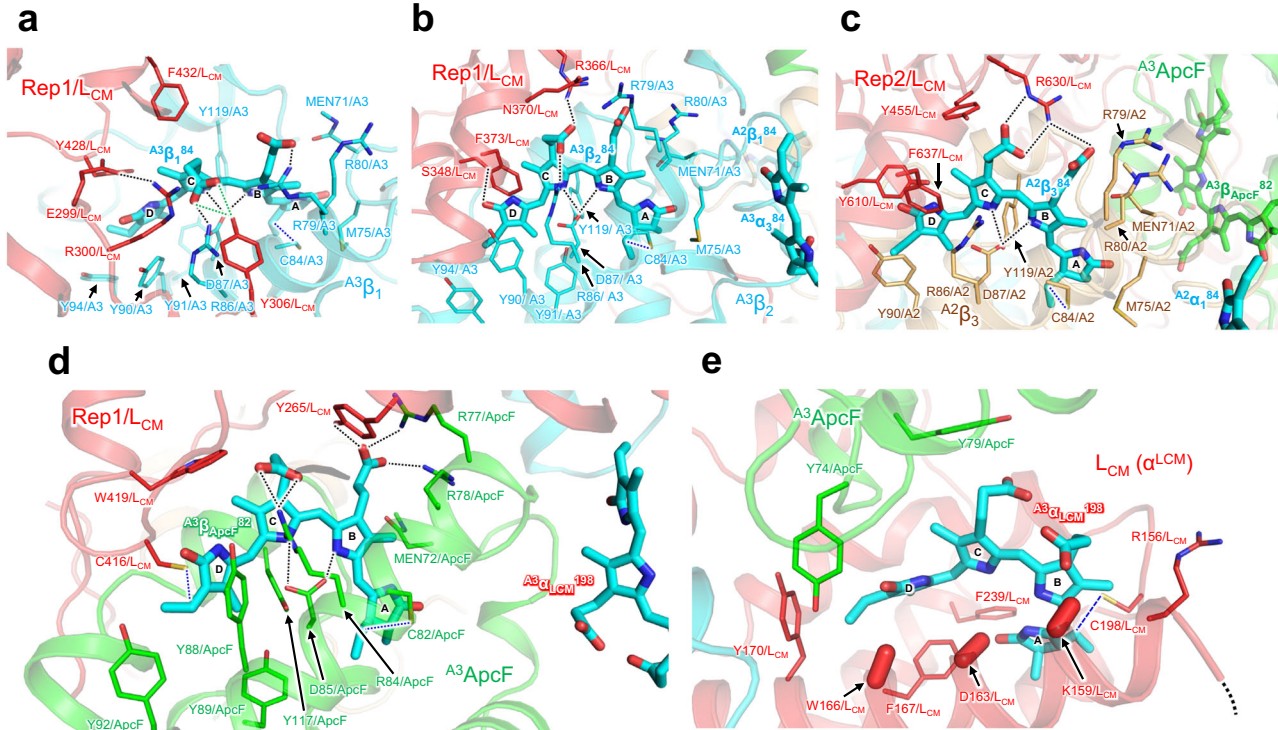

**Fig. 4 Chromophores and their surrounding structures in terminal emitter (ApcE [$L_{CM}$]) and ApcF. a, b** Chromophores in the APC trimer A3 and their interaction with Rep1 of ApcE. **c** Chromophore in the APC trimer A2 and its interaction with Rep2 of ApcE. **d** Chromophore of ApcF and its interaction with Rep1 of ApcE. **e** Chromophore of ApcE and its interaction with ApcF. The dashed lines (black) indicate hydrogen bonds. The dotted lines (blue) indicate covalent bonds between PCB and cysteine residue. Amino acid residues for which the side-chain arrangement cannot be precisely displayed are indicated with capsule-shaped objects. The directions of atoms from Cα to Cβ in the residues are indicated by the capsule-shaped objects.

structures will require a higher-resolution structure of the PBS and/or a supercomplex structure interacting with PBS and PSII. In addition, the amino acid residues proximal to $^{A3}\alpha_{LCM}{}^{198}$ exhibited slight differences among species (Fig. 5). In the *T. vulcanus* PBS, the amino acid residue corresponding to Tyr79/ApcF was Tyr (*G. pacifica* and *P. purpureum*), Leu (*Nostoc* 7120), and Phe (*Synechococcus* 7002) in each species. This residue is located a distance from the chromophore ring that makes it unlikely to form a π–π interaction; however, it may affect the absorption property of the chromophore because polar/charged amino acid residues affect the absorption energy of pigments[40].

**Structure of PC rod and arrangement of chromophores.** The PC rod is formed by PC monomers consisting of α-subunit (CpcA) and β-subunit (CpcB) to form two PC hexamers (Fig. 6a), inside which linker proteins interact. CpcC, CpcD, CpcG1, CpcG2, and CpcG4 have been identified as linker proteins of the PC in *T. vulcanus* PBSs (Supplementary Fig. 2)[26], and we could assign CpcC, CpcD, and CpcG2 in the cryo-EM map of the PC rod (Fig. 6b, c). As CpcG1, CpcG2, and CpcG4 show similar amino acid sequences (Supplementary Fig. 14), CpcG2 assignment in this study is tentative. As observed, the PC rods are dissociated from the prepared PBS, and thus, the obtained cryo-EM map represents the 3D reconstruction using particles combined with multiple CpcGs (CpcG1, CpcG2, and CpcG4). During single-particle analysis, 3D classification can be used to classify foreign and heterogeneous particles; however, the overall structures of CpcG1, CpcG2, and CpcG4 are extremely similar, which made it impossible to classify particles with each CpcG at a 4.2 Å resolution.

In the *Nostoc* 7120 PBS, when three CpcG encoding genes (*CpcG1*, *CpcG2*, and *CpcG4*) were deleted, the PC rods could not bind to the PBS core[7]. When one of the CpcG encoding genes was

deleted, the PC rods could interact with the PBS core but not in the same manner as the wild-type PBS. This suggests that the composition of the CpcG protein is different in each PC rod[7]. The cryo-EM map showed the linker proteins extending from the PC rods interacting with PBS core. Although the quality of the cryo-EM map was insufficient to build a structural model, the high identity of the amino acid sequences suggests that the arrangement of CpcGs of *T. vulcanus* is similar to that of *Nostoc* 7120 (Supplementary Fig. 15)[7,22]. The C-terminal region of the CpcG protein in Rt (Rt') and Rb (Rb') mainly interacts with the peripheral region of the B cylinder and the peripheral region of the A (A') cylinder, respectively. The C-terminal region of CpcG interacts weakly with the cylinders of the PBS core and also supports a rod.

The interaction between PCs and linker proteins in the PC rod is asymmetric (Fig. 6), which is induced by the interactions between CpcC, CpcD, and CpcG2 in the rod. To investigate how the PC rod identified by cryo-EM in this study differs from previously described PC rod structures, we superimposed the PC rod solved by X-ray crystallography[32]. We superposed the Cα backbone of the polypeptide in the PC monomer 1 (consisting of Chain A and Chain B) to the corresponding ones in *T. vulcanus* (3O2C) and *Nostoc* 7120 (7EYD). Then, RMSDs between the other PC monomers were estimated, while maintaining the whole PC rod arrangement. Although there were no major differences in the structure of each PC monomer comprising the PC rods, the interaction of the linker proteins in the PC rods caused a significant shift in the arrangement of Disk B in the PC rod.

The superposition of the PC rod solved by cryo-EM and X-ray crystallography (PDB code: 3O2C; the PC structure from *T. vulcanus*) revealed that the change in the arrangement of PC monomers (monomers 1–12) in the PC rod was primarily caused

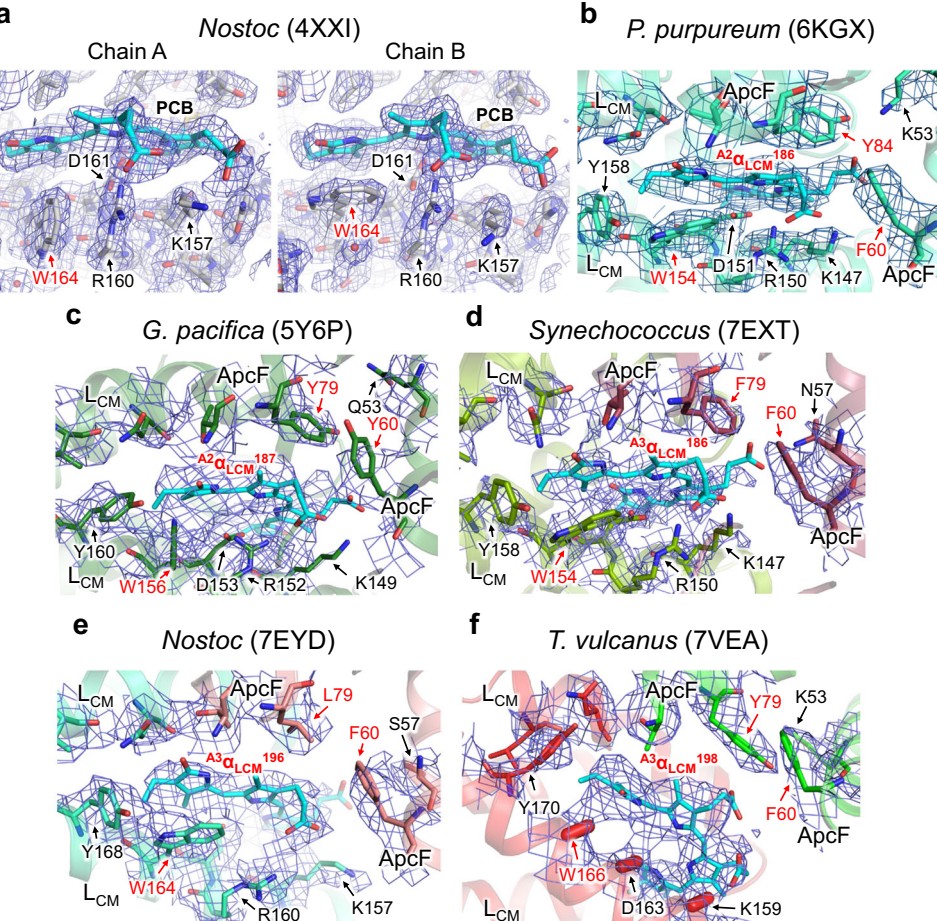

**Fig. 5 Comparison of chromophores and their surrounding structures in the terminal emitters (ApcE [L$_{CM}$]) of each species. a** Crystal structure of *Nostoc* 7120 ApcE at 2.2 Å resolution. **b** Cryo-EM structure of the red algal *P. purpureum* ApcE at 2.8 Å resolution. **c** Cryo-EM structure of the red algal *G. pacifica* ApcE at 3.5 Å resolution. **d** Cryo-EM structure of the cyanobacterial *Synechococcus* 7002 ApcE at 3.5 Å resolution. **e** Cryo-EM structure of the cyanobacterial *Nostoc* ApcE at 3.9 Å resolution. **f** Cryo-EM structure of the cyanobacterial *T. vulcanus* ApcE at 3.7 Å resolution. These ApcE structures were superimposed with their density map contoured at 1σ (**a**), 3σ (**b**), 2σ (**c**), 5σ (**d**), 3σ (**e**), and 3σ (**f**). Amino acid residues for which the side-chain arrangement cannot be precisely displayed are indicated with capsule-shaped objects. The directions of atoms from Cα to Cβ in the residues are indicated by the capsule-shaped objects.

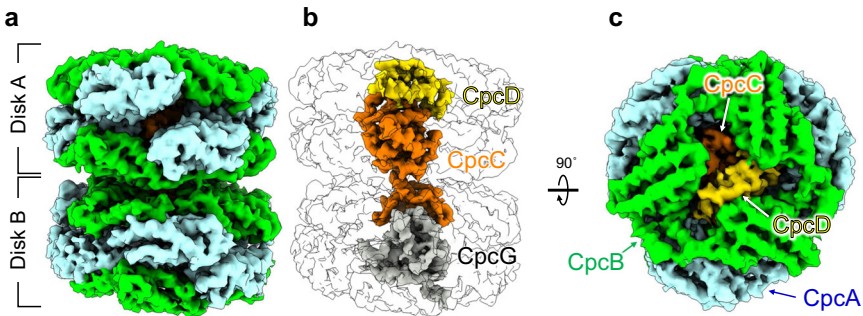

**Fig. 6 Structure of PC rod including linker protein. a** cryo-EM map of the PC rod (Disk A and Disk B). **b** Arrangement of the linker proteins (CpcC, CpcD, and CpcG) in the PC rod. **c** Structure of the PC rod in **b** rotated 90˚.

by the interaction with linker proteins (Fig. 7 and Supplementary Table 7). The RMSD between monomers 1 in Disk A (Fig. 7a) is small (0.55 Å), whereas that between monomers 3 is large (2.60 Å) owing to the interaction between CpcD regions (residues 55–65 [red] and 68–74 [green]) and CpcB regions (residues 109–122 [orange]) in monomer 3 (Fig. 7a). The RMSD between monomer 2 is the largest in Disk A (4.06 Å) (Fig. 7a), suggesting that the change in the arrangement of monomer 3 was caused by

the change in the arrangement of monomer 2, which was required to maintain the interaction between monomers 2 and 3. In Disk A (Fig. 7b), the RMSD between monomer 4, which is located on the lower side of monomer 1, is as small as that of monomers 1 (0.76 Å) (Fig. 7b). However, the RMSDs between monomers 5 and monomers 6 are slightly larger (1.49 Å and 1.40 Å, respectively), suggesting that this is due to the interaction between the CpcC regions (residues 123–127 [cyan] and 197–204

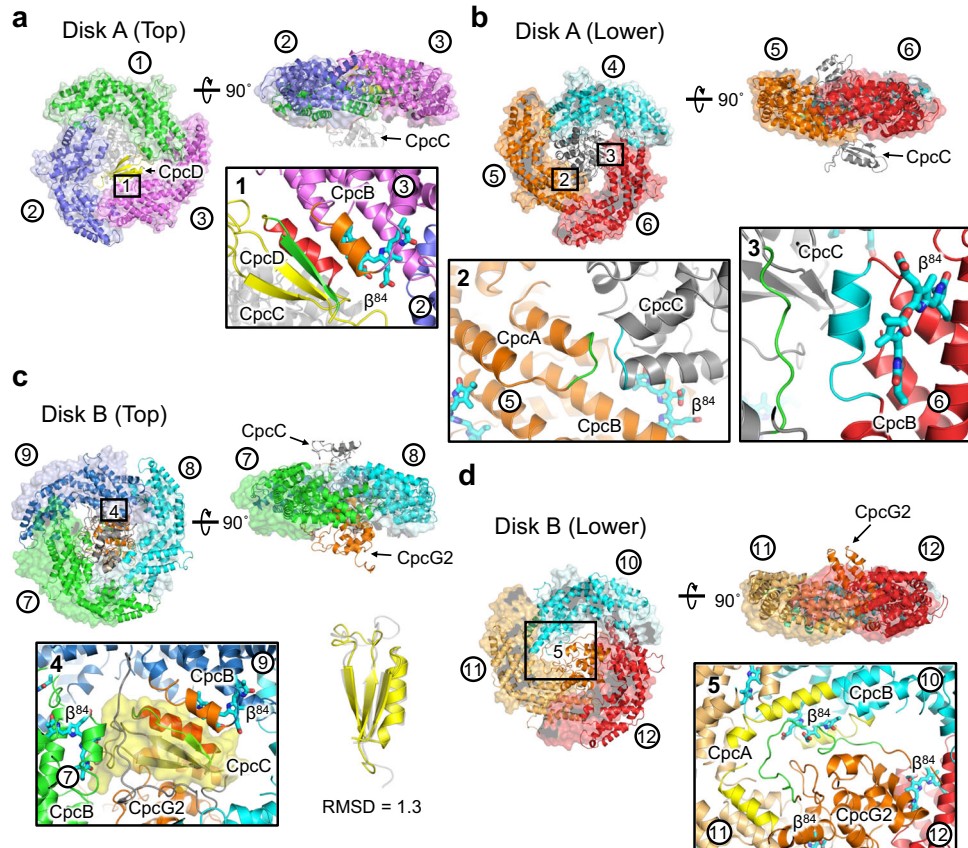

**Fig. 7 Structural comparison of PC rods solved by cryo-EM (PDB: 7VEB) and X-ray crystallography (PDB: 3O2C). a** Superimposition of the top parts of Disk A. The linker protein CpcD (residues 55–65 [red in "panel 1"] and 68–74 [green in "panel 1"]) interacts with a CpcB region (residues 109–122 [orange in "panel 1"]) in PC monomer 3. **b** Superposition of the lower parts of Disk A. The linker protein CpcC (residues 123–127 [cyan in "panel 2"] and 197–204 [green in "panel 3"]) interacts with CpcB regions (residues 14–17 [green in "panel 2"] of PC monomer 5 and 113–122 [cyan in "panel 3"] of PC monomer 6, respectively). **c** Superimposition of the top parts of Disk B. The "CpcD-like structure (residues 234–287)" in CpcC interacts with a CpcB region (residues 109–122, orange in "panel 4") in PC monomer 9. The RMSD of the regions (residues 234–287) in CpcC and CpcD is 1.3. **d** Superposition of the lower parts of Disk B. The linker protein CpcG2 (residues 9–38 [green in "panel 5"]) interacts with regions in monomers 10 (residues 78–90 and 109–122 [yellow in "panel 5"]) and 11 (residues 1–15 and 105–115 [yellow in "panel 5"]). The areas marked "1–5" are the magnified view of the interacting parts of the linker proteins and each PC monomer. Helix and transparent surface models represent PC rods solved by cryo-EM and X-ray crystallography, respectively. Panels 1–6 show magnified views of the interactions of each PC trimer with the linker proteins.

[green]) and those of monomers 5 (residues 14–17 [green]) and 6 (residues 113–122 [cyan]).

Notably, the arrangement of the monomers in Disk B is significantly altered relative to that in the PC rods solved by X-ray crystallography (Fig. 7c, d). This major rearrangement also occurred in the *Nostoc* 7120 PC rod, strongly suggesting that this is due to the interaction of the linker proteins (CpcC and CpcG) in Disk B. The RMSDs between the monomers (7–12) in Disk B are 21.3 Å, 14.8 Å, 10.6 Å, 11.6 Å, 18.3 Å, and 15.4 Å, respectively. Disk B (Fig. 7c) interacts with CpcC and CpcG2, and the "CpcD-like structure (residues 234–287)" in CpcC specifically interacts with the CpcB region (residue 109–122 [orange]) in monomer 9, with this interaction being very similar to that between CpcD and CpcB in monomer 3. For Disk B (Fig. 7d), as in Disk B (Fig. 7c), there was a significant change in the arrangement compared to the crystal structure, namely, interactions between residues in the CpcG2 region (residues 9–38 [green]) and monomers 10 (residues 78–90, 109–122 [yellow]) and 11 (residues 1–15, 105–115 [yellow]) (Fig. 7d). The linker proteins contribute to the structural stabilization of the PC rods, and many regions in the linker proteins are in proximity to chromophores in the rods, suggesting that they also contribute to fine-tuning the energy levels of these chromophores.

In addition, we superimposed the *T. vulcanus* PC rod and *Nostoc* 7120 PC rod (Rs1: PC rod containing CpcG2) for structural comparison (Supplementary Fig. 16 and Supplementary Table 7). The overall structures are very similar, with the arrangement of Disk A and Disk B in the *Nostoc* 7120 PC rod different from that in the crystal structure. This suggests that the symmetric interactions between the PC trimer in the crystal structure are due to crystal packing. One difference between the two PC rods is that the *Nostoc* 7120 PC rod (Rs1) does not contain CpcD, and the arrangement of the PC monomer in the vicinity of CpcD in Disk A differs from that of the *Nostoc* 7120 PC rod (RMSD = 2.79), suggesting that this change in the arrangement was caused by the interaction between CpcD and CpcB in monomer 3 of *T. vulcanus*.

**Possibly energy transfer pathway in PC rod**. The arrangement of chromophores, especially at the boundary between Disks A and B in the PC rod, differs from that inferred from the crystal packing of the PC structure due to the interaction with the linker proteins (Fig. 8). CpcD ($L_R$), a small rod cap linker domain, is the outermost linker protein of the intact PBS, and the excitation energy is transferred from Disk A to Disk B in the PC rod. Similar

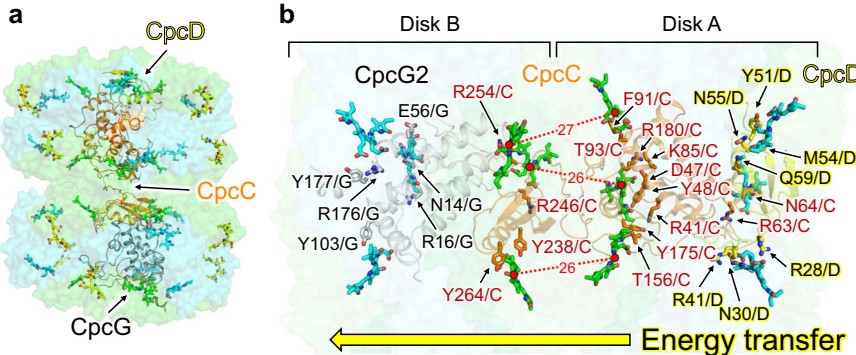

**Fig. 8 Possible energy transfer pathway in the PC rod. a** Arrangement of chromophores in the PC rod. $\alpha^{84}$, $\beta^{84}$, and $\beta^{155}$ are colored cyan, green, and yellow, respectively. **b** Arrangement of $\beta^{84}$s in a PC rod interacting with linker proteins. $\beta^{84}$s at the boundary between Disk A and Disk B are colored green, and these chromophores are involved in the energy transfer between Disk A and Disk B. The amino acid residues of the linker proteins near $\beta^{84}$s are indicated in B.

to the PBS core, excitonic coupling between chromophores contributes to the energy transfer in the PC rods. This interaction results in energy transfer between the PC monomers in the PC trimer, from β155 to β84 and α84 to β84[52]. β84s in Disk A and Disk B, which interact with CpcC, are in close proximity to each other in the PC hexamer of the PC rod, suggesting that the excitation energy is transferred from Disk A to Disk B in the PC rod via the chromophores (β84s) interacting with CpcC (distances between chromophores: 26–27 Å) (Fig. 8). In previous spectroscopic studies of PC, β84 in a PC rod interacting with linker proteins was proposed to be involved in the energy transfer between PC hexamers[52–54]. Therefore, six β84s (Fig. 8b) at the boundary between Disks A and B are likely involved in the energy transfer between the disks. The characteristic amino acid residues around the chromophores in the linker proteins and the interactions between these residues and the chromophores are likely crucial for unidirectional energy transfer in the PC rod.

In summary, cryo-EM analysis of the PBS core and PC rod from *T. vulcanus* (a hemidiscoidal structure in many cyanobacterial species) revealed the chromophore arrangement in the PBS and its surrounding structure. We identified three APC trimers in the A cylinder of *T. vulcanus* PBS, which differed from that observed in PBS of other cyanobacterial species. The reason for this remains unclear, and thus, detailed investigations are required in the future. The C cylinder and its surrounding structures in *T. vulcanus* PBS and the overall structure of ApcE, one of the terminal emitters, were very similar to those of *Nostoc* 7120 PBS. In particular, the interaction between the chromophore in the ApcE and Trp is highly conserved, and this interaction is crucial for the energy transfer from PBS to PSII. However, the amino acid residues around the chromophore differ slightly among species, suggesting that this reflects the differences in fine-tuning of the energy level of the chromophore among species. The overall structure of the PC rods analyzed by cryo-EM is different from the previously reported structures of PC rods (particularly the interaction between the two disks), which is considered to be due to the interaction with the linker proteins. Because PC rods are easily dissociated from the PBS core, it is difficult to clarify the detailed structure. Therefore, it is necessary not only to analyze the structure of the intact PBS but also to further analyze the dissociated PC rods. Because the structures of PBSs vary among species, information about the PBS structures provides a basis for the studies on the evolution of photosynthesis and diversity of its pathways. In the future, we will perform mutational, spectroscopic, and theoretical studies based on this structural information, which will help deepen our understanding of the function of PBSs.

## Methods

**Preparation of intact PBSs, PBS cores, and PC rods**. Cells of the thermophilic cyanobacteria *T. vulcanus* NIES-2134 were cultured in a phosphate medium[55,56]. Intact PBSs and PBS cores were prepared according to the protocols described by ref. [26]. Thylakoid membranes, PSI, and PSII were prepared according to the protocols described by refs. [55,56]. For PC rods, the intact PBSs were mounted on a holey carbon film-coated copper grid (see "*Sample preparation and data collection for cryo-EM*" for a description of pre-treatment of the grid), and then a small amount of buffer containing 10 mM phosphate buffer was quickly added to reduce the phosphate concentration in the sample to less than 0.5 M. This resulted in the dissociation of the PC rods from the intact PBSs.

**Data collection and processing for negative-staining EM**. The intact PBSs were stained with 2% (w/v) uranyl acetate on a carbon-coated grid for 1 min at 23 °C, and the solution remaining on the grid was removed before drying. The resultant grids were transferred into an electron microscope (JM-2100; JEOL, Tokyo, Japan) operated at 200 kV. Images were recorded using a Oneview camera (AMETEK, Berwyn, PA, USA) at a nominal magnification of 60,000× (pixel size: 1.90 Å). Data processing of the negative-staining EM images was performed using RELION-3.1.0[57]. Contrast transfer function (CTF) parameters were estimated using CTFFIND4-1.10[58] and the intact PBS particles were picked using Xmipp3[59], a reference-free particle picking program. In all, 9,358 particles were picked from 249 images and subjected to the reference-free 2D classification using RELION-3.1.0.

**Sample preparation and data collection for cryo-EM**. To clarify the structure of the PBS core and PC rod, grids for cryo-EM were prepared in the following manner. A holey carbon film-coated copper grid (Quantifoil R1.2/1.3 Cu 200 mesh; Microtools GmbH, Berlin, Germany) that had been pretreated by Au sputtering was glow-discharged for 10 s using an ion coater (JEC-3000FC; JEOL, Tokyo, Japan). For the PBS core, 3.0 μL of the PBS core was applied to the grid and blotted with filter paper for 4 s, then immediately plunge-frozen in cooled ethane using a FEI Vitrobot Mark IV (Thermo Fisher Scientific, Waltham, MA, USA) under 100% humidity at 4 °C. For the PC rod, 3.5 μL of the purified sample was applied to the grid and diluted with 1.5 μL of 10 mM phosphate buffer on the grid. Then, the grid was blotted and manually plunge-frozen in cooled ethane using a homemade plunger. The grids were then introduced into a CRYO ARM 300 electron microscope (JEOL) equipped with a cold-field emission gun and an in-column energy filter with a slit width of 20 eV. Dose-fractionated images were recorded using a K2 summit direct electron detector (AMETEK, Berwyn, PA, USA) in counting mode. All images were corrected using a JEOL automatic data acquisition system[60] with a nominal magnification of 40,000×, which corresponded to a pixel size of 1.24 Å. Nominal defocus ranges and dose rates for PBS core and PC rod were −0.5 μm to −1.5 μm and 84.1 e⁻ Å⁻² with 50 frames and −0.5 μm to −1.5 μm and 50.5 e⁻ Å⁻² with 30 frames, respectively. In total, we collected 4600 and 2865 movies for the PBS core and PC rod, respectively.

**Data processing for cryo-EM**. The collected movie stacks of the PBS core were divided into eight optics groups to correct changes in the beam tilt over time. Drift correction and dose-weighted frame summing were performed using MotionCor2-1.3.2[61], and CTF parameters were estimated using CTFFIND4-1.10[58]. The images were selected for further data processing based on the Thon ring patterns. The PBS core particles were manually picked and subjected to reference-free, 2D classification by RELION-3.1.0[57] to create reference images for automatic particle picking. In all, 128,676 particles were picked automatically and extracted with a pixel size of 2.48 Å. Good averaged classes containing 45,427 particles after 2D classification were subjected to ab initio three-dimensional (3D) reconstruction in cryoSPARC-

2.12.0[62]. Following the 3D classification in RELION, a well-aligned 3D class containing 25,532 particles was extracted with a pixel size of 1.24 Å and used for further 3D refinement with two-fold symmetry enforcement. Post-processing in RELION yielded a 4.75-Å resolution map based on the gold standard FSC. Finally, the resolution reached 3.71 Å after two rounds of Bayesian polishing and CTF refinement.

For PC12mer, all movie stacks were processed using MotionCor2-1.2.1 and Gctf-1.06[63] for drift correction, frame summation, and CTF parameter estimation. The 812,327 particles were picked by convolutional neural network picking using EMAN-2.3.1[64] and binned to a pixel size of 4.96 Å during extraction using RELION-3.0. After two rounds of 2D classification, 309,291 particles were selected and subjected to ab initio model construction using *cis*TEM-1.0.0 beta[65]. Good particles were automatically picked again with a 3D reference using the EMAN-2.3.1. In all, 159,537 particles with a pixel size of 1.86 Å were selected from 1,022,349 picked particles based on 2D and 3D classifications. 3D refinement using RELION-3.0 was performed, and following Bayesian polishing, CTF refinement and additional 3D classification improved the resolution to 4.19 Å with 111,054 particles without any symmetry enforcement. For further details, see Supplementary Figs. 4, 5 and Supplementary Table 1.

**Model building, refinement, and validation**. Initial models of the PBS core and PC rod from *T. vulcanus* were built using reference models of PC and APC (PDB codes: 3O18 and 3DBJ) and homology modeling was performed on the SWISS-MODEL server (https://swissmodel.expasy.org/) by referring to the structures of the linker proteins (PDB codes: 5Y6P and 6KGX). The obtained models were fitted with the cryo-EM maps using the "fit in map" program in UCSF Chimera (version 1.13), and initial models of the PBS core and PC rod were created. The initial models were modified manually using COOT to fit the cryo-EM map and refined using Phenix (version1.19.2). Subsequently, each subunit and its interacting subunits in the PBS core and PC rod were grouped and refined against the cryo-EM map using REFMAC5 (version 5.8.0267) in CCP-EM. Finally, the grouped models were merged into one, and the overall structures (PBS core and PC rod) were refined using Phenix. The refinement statistics of the refined models were obtained using the comprehensive validation program in Phenix. The restraints needed for the ligands in the PBS core and PC rod from *T. vulcanus* were generated by electronic Ligand Bond Builder and Optimization Workbench[66]. The restraint information for a methylated Asn (ligand ID: MEN) and phycocyanobilin (ligand ID: CYC) was obtained from the model of MEN identified in a high-resolution crystal structure (PDB code: 3O18). Comprehensive validation (in Phenix), Q-score[39], and FSC-Q[67] were used to validate the refined structural models (PBS core and PC rod).

**Estimation of the orientation factor between chromophores in the PBS core**. Based on the arrangement of the PCBs and their orientation in the constructed PBS core, an approximate orientation factor, $\kappa^2$, was estimated (Supplementary Table 2). These estimates were made especially for the sites likely to be involved in the energy transfer between cylinders in the PBS core and for the terminal emitters. Excitation energy transfer rate ($k_{EET}$) is given by Eq. (1), where $V$ and $\Theta$ are the electronic coupling factor and the overlap integral, respectively.

$$k_{EET} \propto V^2 \Theta \tag{1}$$

As shown in Supplementary Fig. 11 (a), the transition dipole moment of donor (D) and acceptor (A) are $\mu_D$ and $\mu_A$, respectively. $r$ is the intermolecular center-to-center distance between D and A. $V$ is written by the approximation equation [Eqs. (2) and (3)], and the orientation factor ($\kappa^2$) can be estimated by Eq. (4).

$$V \approx \left[ \frac{\mu_D \cdot \mu_A}{r^3} - \frac{3(\mathbf{r} \cdot \mu_D)(\mathbf{r} \cdot \mu_A)}{r^5} \right] \tag{2}$$

$$\approx \left[ \frac{\mu_D \mu_A \cos\theta_T}{r^3} - \frac{3(\mu_D r \cos\theta_D)(\mu_A r \cos\theta_A)}{r^5} \right] = \frac{\mu_D \mu_A (\cos\theta_T - 3\cos\theta_D \cos\theta_A)}{r^3} \tag{3}$$

$$\kappa^2 = (\cos\theta_T - 3\cos\theta_D \cos\theta_A)^2 \tag{4}$$

The orientation of the transition dipole moment of PCBs was determined by referring to ref. [68] (Supplementary Figs. 11 and 12).

**Absorption spectroscopy**. Absorption spectra of the thylakoid membrane, PSII, PSII, and PBS were measured at 23 °C using a UV-2600 spectrophotometer (Shimadzu, Kyoto, Japan). The spectrum of the thylakoid membrane was measured using the opal-glass method.

**Polypeptide analysis and mass spectrometry (MS)**. Polypeptides denatured with dithiothreitol were separated on a SuperSep gel containing 10–20% acrylamide (Wako, Tokyo, Japan) and then stained with Coomassie Blue (Supplementary Fig. 2c). A gel imaging system (ChemiDoc XRS + system; Bio-Rad, Hercules, CA, USA) was used to photograph the stained gels. Separated polypeptides (a band containing ApcC and CpcD) were identified by MS. The obtained protein bands were treated by in situ digestion using trypsin[69]. The fragmented peptides were analyzed by peptide mass fingerprinting and MS/MS using Autoflex speed (Bruker

Daltonik GmbH, Bremen, Germany). The obtained mass spectra were analyzed by using the MASCOT server (Matrix Science Inc., Boston, MA, USA).

**Phylogenic analysis**. Thirty-eight sequences of ApcE from selected strains of cyanobacteria, glaucophytes, and rhodophytes were obtained by blastp program in the NCBI website (https://blast.ncbi.nlm.nih.gov/Blast.cgi) (Supplementary Figs. 8 and 9). The sequences were aligned using mafft (v.7.478) with the L-INS-i option[70]. The maximum likelihood tree of the ApcE sequences was estimated using iqtree2 (v.2.1.4) with the LG + F + R5 model selected by ModelFinder. The statistical support of the trees was estimated with 1,000 replications of ultrafast bootstrap approximation[71]. Domain organization of ApcE was determined using the hmmscan program in HMMER (v.3.3.2; http://hmmer.org/) with the Pfam database[72]. The phylogenetic tree and domain organization were visualized using iToL (v.4[73]).

**Reporting summary**. Further information on research design is available in the Nature Research Reporting Summary linked to this article.

## Data availability
Atomic coordinates and cryo-EM maps for the reported structure of the PBS core and PC rod from *Thermosynechococcus vulcanus* were deposited in the Protein Data Bank under accession codes 7VEA (PBS core) and 7VEB (PC rod), and in the Electron Microscopy Data Bank under accession codes EMD-31944 (PBS core) and EMD-31945 (PC rod), respectively. The graphs and figures generated in this study are provided in the Supplementary Information/Source Data file. Other data are available from the corresponding authors upon reasonable request. Source data are provided in this paper.

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

## Acknowledgements

We thank research assistants Ms. Yuko Kageyama (Biostructural Mechanism Laboratory, RIKEN SPring-8 Center) and Ms. Rie Uno (Osaka Metropolitan University) for their help with cell cultures, preparation of samples, electrophoresis analysis, and spectroscopy. We also thank Ms. Tomomi Shimonaka at the Graduate School of Science, Osaka City University, for performing MS/MS spectrometry. This work was supported by the Japan Society for the Promotion of Science (JSPS) (JP20H05109 (to KK), JP20K06528 (to K.K.), and JP17H06434 (to N.K.)) and partly by the Joint Usage/Research by Institute of Industrial Nanomaterials, Kumamoto University. JST-Mirai Program Grant Number JPMJMI20G5 (to K.Y.), and the Cyclic Innovation for Clinical Empowerment (CiCLE) from the Japan Agency for Medical Research and Development, AMED (to K.K., T.H., and K.Y.).

## Author contributions

K.K. and N.K. designed the study; K.K. prepared the samples and performed electrophoresis analysis; K.K. performed spectroscopic measurements, K.K. performed the measurement and analysis of the negative-staining EM; T.H. measured EM micrographs, and T.H. and K.K. processed the EM data (PBS core: T.H.; PC rod: K.K.) and reconstructed the final EM map (PBS core: T.H.; PC rod: K.K.); K.K. performed structural analysis; Y.H. performed phylogenetic tree analysis; D.K. commented on the data analyses; K.Y., N.K., and M.M. supervised the project; K.K., T.H., and K.Y. wrote the draft manuscript; K.K., T.H., and K.Y. revised the final manuscript; and all authors contributed to the interpretation of the results and improvement of the manuscript.

## Competing interests

The authors declare no competing interests.
