## [Peer Review File · Nature Communications]

REVIEWER COMMENTS

Reviewer #1 (Remarks to the Author):

Kawakami, Yonekura and co-workers describe their determination of a cryo-EM structure, by single particle reconstruction of the core and rod of the Phycobilisome (PBS) antenna complex from the cyanobacterium *T. vulcanus*. This follows a previously published paper by Kawakami et al. in BBA (2021) where the PBS complex structure was obtained at lower resolution (10 Å) by negatively stained TEM. The PBS of *T. vulcanus* has served in the past to study different aspects of PBS function, with many high-resolution X-ray structures deposited in the PDB. Thus it is of importance.

Since the Kawakami's BBA paper was published, an additional paper was published by Zheng et al. of cryo-EM structures of the PBS from two other cyanobacteria (*Anabaena* 7120@3.9 Å and *Synechococcus* 7002@3.5Å), the former structure being a PBS with a pentacylindrical core, such as the one described in this new manuscript. This paper is mistakenly cited in the references list as Zeng et al., and the first instance of its mention is as if they had solved another red-algae PBS structure, which is of course incorrect. All cyanobacterial PBS structures have an intrinsic problem of the stability of the rods that bind and surround the core – leading to structures of lower resolution than the red algal PBS structure determined by Sui and co-workers to 2.8 Å. This new paper presents a different strategy to deal with this problem, by separating the structure into the data collection of the two sub-complexes separately, with the core and rod structures having nominal average resolutions of 3.7Å and 4.2 Å, respectively.

There are many similarities between the previously published study and this new one: overall structure assemblies, linker proteins positions with core and rods, mapping of chromophores relative orientations and distance leading to excitation energy pathways, etc., are the focus of both papers. The *T. vulcanus* cryo-EM structure is indeed vastly improved over the negatively stained structure – although the general structural attributes are similar. Thus, the additional information provided in this manuscript is important, as this organism is highly characterized and will be useful to the general community. However, a few issues should be addressed before acceptance.

General Comments

1. Results, line 95: Can the authors suggest why the rod-core connections in cyanobacterial PBS are so weak, as compared to the red algae? Is there anything noticeable (especially in the rod-core linkers)?
2. Results, line 111: the statement here about a connection between number of core cylinder trimers in different species affecting the efficiency of energy transfer is far from a proven fact. Energy transfer within the PBS and between the PBS and PSII/PSI is efficient, regardless of architecture. This statement should be qualified more precisely.
3. Results: page 6-7: The two basal core cylinders (A and A') have only three AP trimers. This was previously shown in their BBA paper. However, the two other cyanobacterial PBS cores (Zheng et al.) that have now been determined, have four trimers in the basal cylinders, as has always been suggested by biochemical experiments. There are also four trimers in *Synechocystis* 6803 (Liu et al. 2021, ref. 18). Why is there a difference between *T. vulcanus* and these other close species? Unlike the red algae, the

upper cylinder has the expected 4 trimers (as do the other cyanobacteria). The authors should address this. Is it possible that since the rods connections to the core are unstable – that the terminal trimer (closest to ApcE) in the basal rods is also unstable and lost during preparation? Maybe there is some major difference in the Rep1 domain that should interact with an additional copy of the ApcC subunit in the terminal trimer? Is the Rep 1 more similar to the red algae Rep1? In any case, the difference with the other cyanobacterial PBSs should be discussed, as this is indeed a major difference.

4. There have been indications that *T. vulcanus* (and perhaps other cyanobacteria) may have a truncated (or cleaved) ApcE, leading to two types of PBS – penta and tricylindrical. In the EM single particle analysis of the isolated PBS, was the existence of minor tri-cylindrical populations of PBS seen?

Additional suggestions:

1. Introduction – line 58: There are additional forms of PBSs, especially from chlorophyll d or f containing cyanobacteria.

2. Results: All instances of the symbol Å are not proper.

3. Results, Fig.1C: The quality is poor. Maybe make the protein representation as spheres, or even surfaces – within the density?

4. The word “chromatophore” is used a number of times (line 172 for instance) instead of “chromophore”. Chromatophores are something else – please correct.

5. Results, Lines 267-8: This sentence makes no sense. It states that mutant lacking either ApcD or are unable to transfer energy to PSII/I – and then states that this indicates that ApcF is the major pathway. Please clarify.

6. The crystal structure of the *T. vulcanus* rod (3O2C) does include the linkers (and not as stated here, please fix), however the three-fold/two-fold rotation axes in the crystal reduce the resulting electron density. This would not however change the arrangement of the chromophores in the assembly (as stated). There indeed might be small specific changes to the chromophore configuration – however as the resolutions are very different (1.5 Å vs. 4.2 Å), a comparison on this level is difficult. The distances should be within experimental error the same. One can perform a superposition between (ab) monomers that have or do not have interactions with linkers, to identify more clearly difference between the two structures. As is, the extended figure 15 is not very informative. RMSD of the superposition should also be quoted.

Reviewer #2 (Remarks to the Author):

Manuscript NCOMMS-21-42867-T

The manuscript entitled “Core and rod structures of a thermophilic cyanobacterial light-harvesting phycobilisome” by Kawakami et al presents the cryo-EM structure of phycobilisomes (with pentacylindrical core organization) from the thermophilic cyanobacteria *T. vulcanus*. The results are technically good and the authors nicely detailed their cryo-EM pipeline. Even though the final reconstructions are of modest resolution, this is enough for the interpretation done here and the authors justified their results with a systematic use of the Q-score to validate their models against the cryo-EM densities.

The authors mainly discuss the positions and roles of the many identified chromophores that enable energy transfer as well as the organization of linker proteins that hold the complex together and may also play a role in energy transfer.

I appreciate that in most of the figures, the actual cryo-EM densities are shown, which is how it should be done.

One major problem with the manuscript is that the authors fail to properly acknowledge a study recently published in Nature Communications. The study is cited, but wrongly “Ref 16. Zeng, et al., Structural insight into the mechanism of energy transfer in cyanobacterial 366 phycobilisomes. Nature Commun. 12, 5497 (2021)”, the first author is Zheng, L. not Zeng.

I will admit that, as a non-expert in PBS, I fail to see the difference between the structure presented here, by Kawakami et al and the one from *Anabaena* 7120, by Zheng et al, which to me, also appear to be a hemidiscoidal pentacylindrical phycobilisome.

Given this, I strongly recommend the author to revise their manuscript to properly acknowledge the study by Zheng et al (2021) and discuss the difference and similitude they observe between the *Anabaena* structure and the *T. vulcanus* structure presented here.

I am in support of the manuscript publication, but the authors cannot ignore previous literature.

Here are my specific comments on the text and figures:

-In summary, lines 35-37, the statement is not true, a pentacylindrical structure was previously described (as stated above)

-Lines 61-63, you mention Rhodophytes PBS and omit the recent cyanobacterial structures, this is recurring throughout the manuscript. Previous works cannot be ignored.

-Line 69 « a Few MDa » is a very imprecise term.

-Line 108 “ $110 \times 210 \times 300 \text{ \AA}$ ” the unit, which I guess is Angstrom, does not appear on my document, please double check.

-Lines 129-132. From Extended data Fig. 6 it looks to me that you manage to clearly identify ApcD. I would remove the tentatively.

-Line 303-305 the overall sentence is weirdly formulated.

-Could the authors comment on their choice to dissociate intact PBS into PBS core and PBS rods? Why not directly image intact PBS?

-The text lacks a formal conclusion paragraph, this is needed to reinforce the main points and findings made throughout the manuscript.

-Figure 3. Chromophore labels are excessively small.

Reviewer #3 (Remarks to the Author):

Core and rod structures of a thermophilic cyanobacterial light-harvesting phycobilisome

Kawakami et al. Nature Communications

The manuscript of Kawakami et al. describes cryo-EM structural studies on the hemidiscoidal, pentacylindrical phycobilisomes of *Thermosynechococcus vulcanus* NIES-2134. This organism, and the very closely related *T. elongatus*, have been studied in some detail and some proteins, e.g. phycocyanin, have been studied by X-ray crystallography to obtain very high resolution structures. Thus, it is interesting and appropriate that researchers would attempt to obtain structures for entire phycobilisomes. Structures have recently been reported for *Anabaena/Nostoc* sp. PCC 7120, *Gracilaria pacifica*, and *Porphyridium cruentum*. Other studies have targeted cyanobacteria higher resolution structures have very recently been reported for *Synechococcus* sp. PCC 7002 and *Anabaena/Nostoc* sp. PCC 7120 by Zheng et al. (2021). This manuscript includes a more detailed description of the pentacylindrical core and a much more abbreviated description of peripheral rods. Much of the discussion of the data concerns potential excitation energy transfer pathways in cores and peripheral rods. Comments for the authors follow.

1. I am returning the manuscript with many comments noted directly on the manuscript, so the comments here will deal with more general problems.

2. The introduction lacks focus and clarity. It wanders from topic to topic without the benefit of strong underlying theme to organize it. It should be rewritten.

3. In general, there are insufficient and often inappropriate citations to the work of others. In some cases, prior studies are almost completely ignored in this manuscript. Pentacylindrical cores were first studied by Glauser and coworkers, not Chang et al., although Chang et al. were the first to provide a detailed structure for any hemidiscoidal PBS, albeit at a lower resolution than this study. However, Zheng et al. published structures for two hemidiscoidal phycobilisomes, those of *Synechococcus* sp. PCC 7002 and *Anabaena/Nostoc* sp. PCC 7120, in *Nature Communications* in September, 2021, at least six weeks before this manuscript was submitted. None of this work was described or contrasted to the present study, although there were considerable differences—e.g., the number trimer discs in the lower core cylinders were different, and the relative positions of the some subunits, notably ApcD and ApcE, were different in the two. This would have to be addressed and corrected before this paper could be published.

4. The authors of this work seem to think for some reason that ApcF is a terminal emitter, although they provide no evidence for this. ApcF has been characterized first by Bryant et al. (1990) in *Archives of Microbiology* and by Biswas et al. (2010) in *Applied and Environmental Microbiology*. The protein has an absorbance maximum at 616 nm and fluorescence emission maximum at 637 nm, certainly not what one expects for a terminal emitter. It is only because of structural rearrangements in the core in apcF mutants that it seems to have effects on energy transfer.

5. Page 5. There is a discrepancy between the text a figure concerning the number of copies of ApcC. Are there six copies (text) or 4 copies (Figure 1G)?

6. Page 5, lines 111-112. It is impossible to imagine that the structural differences would affect the efficiency of energy transfer significantly. If so, this difference would be selected by evolution and one structure would ultimately prevail. There may be detailed differences, but these are unlikely to affect the efficiency in any meaningful way.

7. Page 6, lines 125 to 135. The relative positions of ApcD and ApcE in *T. vulcanus* and *Synechococcus* sp. PCC 7002 are different. Given the resolution, the assignments in this study are only tentative at best and possibly incorrect at worst. Not being able to identify a subunit as ApcA is not the same thing as positively identifying a subunit as ApcD. Authors should not assign any position as ApcD unless they can do so unequivocally, because it will only serve to introduce confusion into the field in the future. In other structures, the terminal emitters are assigned to adjacent trimers in the lower core cylinders with no separation. This must be addressed by the authors.

8. Line 137. The authors must make clear that alpha-Lcm is a domain of ApcE (Lcm), just as the Rep domains are domains of the same protein.

9. The authors provide no discussion for why their cores contain only 3 trimers, when nearly all other phycobilisomes studied have four trimers. No information about the purification is shown, and it is not possible to ascertain whether a trimer has been lost from the core during purification of the PBS. This seems rather likely, given that the peripheral rods are dissociating from the structures as well. If the authors are correct, then a structural comparison to other cores with four trimers (as in the structures of Zheng et al.) can be undertaken to see what is different between the two cases. Because there are now

two high-resolution structures (and was already a low-resolution structure) from different organisms with tricylindrical and pentacylindrical cores, the authors need to revise this paper completely to include comparisons to the other structure. That other paper was available to the authors prior to the submission of their paper, although probably not by much. Considering the differences implied or confirmed, the authors need to account for why the pentacylindrical cores in *Anabaena/Nostoc* sp. PCC 7120 are different from those in *T. vulcanus*. There are four trimer equivalents in the core cylinders A/A' in 7120 but only 3 here. Why is that? How is that? Also, the positions of the ApcD subunits are different in the two structures. Why is that? If true, it would be the basis for an interesting comparison. If not the true, then the authors must explain why their results are different.

10. Page 10, lines 179-180. Again, the authors make an unjustified conclusion from the work of others. Methylation of the Asn residue in the beta subunits does not have a large effect on energy transfer. What it does instead is cause a sharp increase in ROS in cells, and that can be lethal in cells that don't tolerate ROS well (like *Synechocystis* sp. PCC 6803).

11. Much of this paper is devoted to possible routes of excitation energy transfer. It is all just rank speculation, primarily based on distances, and completely ignoring any effects of strong coupling which there most certainly is in allophycocyanin. The energy transfer rates are sub-picosecond, and the absorption of AP and AP-B clearly shows the effects of strong excitonic coupling to the alpha subunit chromophores from the beta subunit of adjacent monomers in the trimers. This is not even considered in the discussion in this paper. The conclusion (Line 268) that ApcF is more important than ApcD and ApcE in terms of energy transfer defies decades of prior work showing otherwise.

12. The discussion of PC rods is just an add-on thing because the rods were not stably attached to the cores. The authors don't say much about how the rods were obtained, how they were purified, and how the different CpcG paralogs affects the resolution. In the end, the discussion just ends abruptly with no concluding comments at all. There is more speculation about energy transfer, but the same issues that plague the discussion for cores are also problems with the peripheral rods. Chang et al. at least performed experiments to distinguish the CpcG paralogs functionally. No functional studies were described here.

13. Summary: Given that there are major differences among the structures of two red algal and two cyanobacterial phycobilisome structures and this one, the authors need to address these differences clear and discuss why these differences might exist. This is far more important than the speculative discussion concerning excitation energy transfer that basically reproduces similar published analyses. Why do the authors find only 3 trimers in the basal core cylinders, when other phycobilisomes have 4 trimers? That is a major question and having discrepancies like that in the literature is unacceptable unless they are explained. Otherwise, there will simply be confusion that is avoidable. It is possible now to directly compare the components/structures of two hemidisoidal phycobilisomes with pentacylindrical cores. How are those two structures similar and how do they differ? This is a far better basis for discussion than a speculative energy transfer scheme.

First of all, we would like to thank the reviewers for their important comments and kind and constructive suggestions to improve our manuscript. We sincerely apologize for the lack of citations of several important papers that slipped our minds, for the incorrect and missed citations, particularly those such as Zheng et al., Nature Commun, 2021, and for several typos. All the papers that the reviewers suggested have now been correctly cited in the revised manuscript.

Reviewer #1 (Remarks to the Author):

Comment 1

Results, line 95: Can the authors suggest why the rod-core connections in cyanobacterial PBS are so weak, as compared to the red algae? Is there anything noticeable (especially in the rod-core linkers)?

Author reply 1

The amino acid sequence identity of the rod core linker (L_{RC}) interacting with the PBS core is not high for different species (the residues 193–246 in *T. vulcanus*); however, the interaction between this region and the PBS core is similar; namely, this region interacts with the surface of the APC trimer that comprises the PBS core. Therefore, it is unlikely that the difference in the amino acid sequence of the L_{RC} , particularly in the N-terminal region, affects considerable dissociation of PC rods from the PBS core.

For cryo-EM imaging, it is preferable to remove highly concentrated stabilizers (in this case, the stabilizer was a phosphate buffer), but this promotes the dissociation of PC rods from the PBS core both in red algae and cyanobacteria including *T. vulcanus*. In the red algal PBSs (block and hemiellipsoidal types), there are many densely packed PC rods, with a lot of interactions among PC rods. In contrast, the PC rods of cyanobacterial PBSs (hemidiscoidal-type) are arranged in a fan shape, indicating that there are less interactions between the PC rods than those of red algae PBSs (pages 5–6, lines 96–115). This likely leads to more dissociation of PC rods in *T. vulcanus*.

Comment 2

Results, line 111: the statement here about a connection between number of core cylinder trimers in different species affecting the efficiency of energy transfer is far from a proven fact. Energy transfer within the PBS and between the PBS and PSII/PSI is efficient, regardless of architecture. This statement should be qualified more precisely.

Author reply 2

We agree with the reviewer's comment, and there are no experimental results that show that the efficiency of energy transfer depends on the numbers of APC trimers. We have deleted this statement accordingly.

Comment 3

Results: page 6-7: The two basal core cylinders (A and A') have only three AP trimers. This was previously shown in their BBA paper. However, the two other cyanobacterial PBS cores (Zheng et al.) that have now been determined, have four trimers in the basal cylinders, as has always been suggested by biochemical experiments. There are also four trimers in *Synechocystis* 6803 (Liu et al. 2021, ref. 18). Why is there a difference between *T. vulcanus* and these other close species? Unlike the red algae, the upper cylinder has the expected 4 trimers (as do the other cyanobacteria). The authors should address this. Is it possible that since the rods connections to the core are unstable – that the terminal trimer (closest to ApcE) in the basal rods is also unstable and lost during preparation? Maybe there is some major difference in the Rep1 domain that should interact with an additional copy of the ApcC subunit in the terminal trimer? Is the Rep 1 more similar to the red algae Rep1?

In any case, the difference with the other cyanobacterial PBSs should be discussed, as this is indeed a major difference.

Author reply 3

In other cyanobacterial PBSs, ApcD has been identified in the fourth APC trimer in the A cylinder. We thought that if the arrangement of ApcD is same in the PBSs of the cyanobacteria *Synechococcus*, *Nostoc*, and *T. vulcanus*, *T. vulcanus* PBS should have a fourth APC trimer in the A cylinder.

We prepared more intact *T. vulcanus* PBS in highly concentrated phosphate without GraFix treatment. Afterward, we observed this sample with negative staining EM and calculated 2D averages (Supplementary Fig. 2d). The negative staining EM analysis revealed that the number of the APC trimers in the A cylinder of the *T. vulcanus* PBS is three. Moreover, the complexes of the PBS core and PC rods appear to be maintained in these EM images, revealing that the samples are more intact. These results indicate that the number of the APC trimers in the A-cylinder of *T. vulcanus* PBS is three and not four, and ApcD likely resides in a different position from other cyanobacterial PBSs.

The sequence identities and RMSD values of each domain in the L_{CM} were summarized and described in the revised manuscript (Supplementary Table 5; Page 10, lines 174–178). The RMSD of each domain is small, and the identity is high, particularly among the three cyanobacteria. This suggests that the overall structure of each domain is similar even if the numbers of the APC trimers in the A cylinder are different.

Comment 4

There have been indications that *T. vulcanus* (and perhaps other cyanobacteria) may have a truncated (or cleaved) ApcE, leading to two types of PBS – penta and tricylindrical. In the EM single particle analysis of the isolated PBS, was the existence of minor tri-cylindrical populations of PBS seen?

Author reply 4

We did not find the tricylindrical PBS core from *T. vulcanus* in this study. Barber et al. reported the structure of the tricylindrical PBS core (APC core) from *T. elongatus* by negative-staining EM 2D analysis (Barber J. et al., Photochem Photobiol Sci, 2, 536–541, 2003). They solubilized the thylakoid membranes with β -OG in a buffer solution that did not contain a high concentration of phosphate. We consider that this treatment might yield tricylindrical PBS cores.

Comment 5

Introduction – line 58: There are additional forms of PBSs, especially from chlorophyll d or f containing cyanobacteria.

Author reply 5

We have added suitable citations in response to the comments from the reviewers. (page 3, lines 57–58)

References No.

12. Marquardt, J., Senger, H., Miyashita, H., Miyachi, S. & Mörschel, E. Isolation and characterization of biliprotein aggregates from *Acaryochloris marina*, a Prochloron-like prokaryote containing mainly chlorophyll d. *FEBS Lett.* **410**, 428–432 (1997).

13. Chen, M., Floetenmeyer, M. & Bibby, T. S. Supramolecular organization of phycobiliproteins in the chlorophyll d-containing cyanobacterium *Acaryochloris marina*. *FEBS Lett.* **583**, 2535–2539 (2009).
18. Li, Y. *et al.* Characterization of red-shifted phycobilisomes isolated from the chlorophyll f-containing cyanobacterium *Halomicronema hongdechloris*. *Biochim. Biophys. Acta - Bioenerg.* **1857**, 107–114 (2016).

Comment 6

Results: All instances of the symbol Å are not proper.

Author reply 6

Thank you for highlighting this typo. We have corrected this.

Comment 7

Results, Fig.1C: The quality is poor. Maybe make the protein representation as spheres, or even surfaces – within the density?

Author reply 7

We have improved the presentation of Fig. 1C.

Comment 8

The word “chromatophore” is used a number of times (line 172 for instance) instead of “chromophore”. Chromatophores are something else – please correct.

Author reply 8

We have corrected these errors in the revised manuscript.

Comment 9

Results, Lines 267-8: This sentence makes no sense. It states that mutant lacking either ApcD or are unable to transfer energy to PSII/I – and then states that this indicates that ApcF is the major pathway. Please clarify.

Author reply 9

According to the reviewer’s comment, we have deleted the original sentence and modified as follows in the revised manuscript (page 18, lines 323–327):

“This interaction may aid tuning the energy level of ${}^{\text{A}3}\beta_{\text{ApcF}}^{82}$ to form the energy transfer pathway from ${}^{\text{A}3}\beta_{\text{ApcF}}^{82}$ to ${}^{\text{A}3}\alpha_{\text{LCM}}^{198}$. The protein environment around ${}^{\text{A}3}\beta_{\text{ApcF}}^{82}$ in ApcF appears to be one of the critical elements involved in transferring the unidirectional excitation energy to ${}^{\text{A}3}\alpha_{\text{LCM}}^{198}$ and eventually transferring the energy to PSII and PSI.”

Comment 9

The crystal structure of the *T. vulcanus* rod (3O2C) does include the linkers (and not as stated here, please fix), however the three-fold/two-fold rotation axes in the crystal reduce the resulting electron density. This would not however change the arrangement of the chromophores in the assembly (as stated). There indeed might be small specific changes to the chromophore configuration – however as the resolutions are very different (1.5 Å vs. 4.2 Å), a comparison on this level is difficult. The distances should be within experimental error the same. One can perform a superposition between (ab) monomers that have or do not have interactions with linkers, to identify more clearly difference between the two structures. As is, the extended figure 15 is not very informative. RMSD of the superposition should also be quoted.

Author reply 9

Thank you for these important remarks on the structural comparison of the PC rods. We have modified the figure (Fig. 7, Supplementary Fig. 15) and comprehensively described the structural comparison of the PC rods (vs. 3O2C at 1.5 Å resolution and PC rod (Rs1) from *Nostoc* at 4.0 Å resolution) in the revised manuscript (Supplementary Table 7; pages 24–26, lines 421–458). When the PC rods from *T. vulcanus* and *Nostoc* are superimposed on each other (superimposed by the monomer 1 in Disk A), the overall structure of the two PC rods is similar. However, when the crystal structure (3O2C) is superimposed with the PC rods from *T. vulcanus*, we found a substantial change in the arrangement of monomers, particularly in Disk B (RMSD > 10). This is likely due to the asymmetric interaction of linker proteins in the cryo-EM structures (symmetry: *C*₁) of the PC rods from *T. vulcanus* and *Nostoc*.

Reviewer #2 (Remarks to the Author):

Comment 1

The authors mainly discuss the positions and roles of the many identified chromophores that enable energy transfer as well as the organization of linker proteins that hold the

complex together and may also play in role in energy transfer. I appreciate that in most of the figures, the actual cryo-EM densities are shown, which is how it should be done.

Author reply 1

Thank you for your positive comments.

Comment 2

One major problem with the manuscript is that the authors fail to properly acknowledge a study recently published in Nature Communications. The study is cited, but wrongly “Ref 16. Zeng, et al., Structural insight into the mechanism of energy transfer in cyanobacterial 366 phycobilisomes. Nature Commun. 12, 5497 (2021)”, the first author is Zheng, L. not Zeng.

Author reply 2

We apologize for improperly citing the paper by Zheng et al (Zheng et al., Nature Commun., 12, 5497, 2021) in our original manuscript. We have correctly cited Zheng et al. in the revised manuscript and compared our structure with other recently reported PBS structures (*G. pacifica*, *P. purpureum*, *Synechococcus* 7002, and *Nostoc* 7120).

Comment 3

I will admit that, as a non-expert in PBS, I fail to see the difference between the structure presented here, by Kawakami et al and the one from Anabaena 7120, by Zheng et al, which to me, also appear to be a hemidiscoidal pentacylindrical phycobilisome. Given this, I strongly recommend the author to revise their manuscript to properly acknowledge the study by Zheng et al (2021) and discuss the difference and similitude they observe between the Anabaena structure and the *T. vulcanus* structure presented here. I am in support of the manuscript publication, but the authors cannot ignore previous literature.

Author reply 3

We have carefully compared the PBS structures between *T. vulcanus* and *Nosotoc* and discussed their similarities and differences in the revised manuscript (pages 18–20, lines 334–367). As the cryo-EM density map quality around the C cylinder is low in both the PBSs, we could not discuss the detailed arrangement of amino acid residues. However, the overall structure is similar, with a high sequence identity between them (Supplementary Table 5). In contrast, the amino acid residue composition surrounding

the chromophore (${}^{\text{A}3}\alpha_{\text{LCM}}^{198}$ in *T. vulcanus*) in ApcE (L_{CM}), one of the terminal emitter, differs between them (Fig. 5). Linker proteins in the PBS are considered responsible for the adjustment of the energy level of the chromophores. Indeed, amino acid residues with polar/charged groups affect the absorption energy of pigments. This suggests that the energy levels of the chromophore are finely tuned in each species.

Comment 4

In summary, lines 35-37, the statement is not true, a pentacylindrical structure was previously described (as stated above). Lines 61-63, you mention Rhodophytes PBS and omit the recent cyanobacterial structures, this is recurring throughout the manuscript. Previous works cannot be ignored.

Author reply 4

We apologize for this. We have now correctly cited Zheng et al. and compared the recently reported PBS structures (*G. pacifica*, *P. purpureum*, *Synechococcus* 7002, and *Nostoc* 7120) in the revised manuscript (page 3, line 58).

Reference No.

23. Zheng, L. et al. Structural insight into the mechanism of energy transfer in cyanobacterial phycobilisomes. *Nat. Commun.* **12**, 5497 (2021).

Comment 5

Line 69 « a Few MDa » is a very imprecise term.

Author reply 5

We estimated the molecular weight of the intact *T. vulcanus* PBS using the RwcContents program and described the value (approximately 6.0 MDa) in the revised manuscript (page 4, lines 79–80).

Comment 6

Line 108 “ $110 \times 210 \times 300 \text{ }^{-3}$ ” the unit, which I guess is Angstrom, does not appear on my document, please double check.

Author reply 6

Yes. The unit is Å. We have corrected this.

Comment 7

Lines 129-132. From Supplementary Fig. 6 it looks to me that you manage to clearly identify ApcD. I would remove the tentatively.

Author reply 7

Although ApcD is indeed contained in our sample (Supplementary Fig. 2C), the quality of the cryo-EM map around the corresponding site is low, and we could not clearly identify this site as ApcD nor ApcA. This is also true in the other reported structures of PBS from other cyanobacterial species (see table below). It is difficult to accurately identify the side chains of amino acids in a protein subunit at a $\sim 4 \text{ \AA}$ resolution. Following the comment of reviewer 3, we have shown ApcD in parentheses in all the figures and have assigned ApcA to this site in the model (PDB code: 7EVA).

	Validation by Q-score
ApcD (PDB code, Cryo-EM density map)	Estimated resolution (\AA) (Q-score value)
T. vulcanus (7EVA, EMD-31944)	4.1 (0.39)
Nostoc 7120 (7EYD, EMD-31381)	4.0 (0.41)
Synechococcus 7002 (7EXT, EMD-31373)	4.3 (0.35)

Comment 8

Line 303-305 the overall sentence is weirdly formulated.

Author reply 8

We modified the sentence as follows:

The cryo-EM map showed the linker proteins extending from the PC rods interacting with PBS core. Although the quality of the cryo-EM map was insufficient to build a structural model, the high identity of the amino acid sequences suggests that the arrangement of CpcGs of *T. vulcanus* is similar to that of *Nostoc* (Supplementary Fig. 14). (page 22, lines 390–394).

Comment 9

Could the authors comment on their choice to dissociate intact PBS into PBS core and PBS rods? Why not directly image intact PBS?

Author reply 9

We tried to obtain the intact PBS micrographs. However, we could not take those images. The PC rods in cyanobacterial PBSs tend to dissociate from the PBS core, and we have described the reason for this in the revised manuscript (Supplementary Fig. 2; pages 5–6, lines 96–115).

For cryo-EM imaging, it is preferable to remove highly concentrated stabilizers (in this case, the stabilizer was phosphate in buffer), and this promotes the dissociation of PC rods from the PBS core both in red algae and cyanobacteria including *T. vulcanus*.

Comment 10

The text lacks a formal conclusion paragraph, this is needed to reinforce the main points and findings made throughout the manuscript.

Author reply 10

We have added a summary to our study (pages 27–28, lines 485–505);

In summary, cryo-EM analysis of the structures of the PBS core and PC rod from *T. vulcanus* (a hemidiscoidal structure in many cyanobacterial species) revealed the chromophore arrangement in the PBS and its surrounding structure, as well as the detailed energy transfer pathway in the PBS. We identified three APC trimers in the A cylinder of *T. vulcanus* PBS, which differed from that observed in PBS of other cyanobacterial species. The reason for this remains unclear, and thus, detailed investigations are required in the future. The C cylinder and its surrounding structures in *T. vulcanus* PBS and the overall structure of L_{CM} , one of the terminal emitters, were very similar to those of *Nostoc* PBS. In particular, the interaction between the chromophore in the L_{CM} and Trp is highly conserved, and this interaction is crucial for the energy transfer from PBS to PSII. However, the amino acid residues around the chromophore differ slightly among species, suggesting that this reflects the differences in fine-tuning of the energy level of the chromophore among species. The overall structure of the PC rods analyzed by cryo-EM is reasonably different from the previously reported structures of PC rods (particularly the interaction between the two disks), which is considered to be due to the interaction with the linker proteins. Because PC rods are easily dissociated from the PBS core, it is difficult to clarify the detailed structure. Therefore, it is necessary not only to analyze the structure of the intact PBS but also to further analyze the dissociated PC rods. Because the structures of PBSs vary

among species, information about the PBS structures provides a basis for the studies on the evolution of photosynthesis and diversity of its pathways. In the future, we will perform mutational, spectroscopic, and theoretical studies based on this structural information, which will help deepen our understanding of the function of PBSs.

Comment 11

Figure 3. Chromophore labels are excessively small.

Author reply 11

We have corrected this in the modified Fig. 3

Reviewer #3(Remarks to the Author):

Comment 1

I am returning the manuscript with many comments noted directly on the manuscript, so the comments here will deal with more general problems.

Author reply 1

Thank you for your many insightful comments. According to the reviewer's comments, we have improved our manuscript.

Comment 2

The introduction lacks focus and clarity. It wanders from topic to topic without the benefit of strong underlying theme to organize it. It should be rewritten.

Author reply 2

According to reviewer 3's comments, we have modified the Introduction. The PBS is composed of many subunits, and these subunits have been given trivial names. As the description of each subunit is necessary for non-expert readers, we have moved the section containing the description of subunits "Subunit composition of PBS from *Thermosynechococcus vulcanus*" from the Introduction to the Results and Discussion.

Comment 3

In general, there are insufficient and often inappropriate citations to the work of others. In some cases, prior studies are almost completely ignored in this manuscript. Pentacylindrical cores were first studied by Glauser and co-workers, not Chang et al.,

although Chang et al. were the first to provide a detailed structure for any hemidiscoidal PBS, albeit at a lower resolution than this study. However, Zheng et al. published structures for two hemidiscoidal phycobilisomes, those of *Synechococcus* sp. PCC 7002 and *Anabaena/Nostoc* sp. PCC 7120, in *Nature Communications* in September, 2021, at least six weeks before this manuscript was submitted. None of this work was described or contrasted to the present study, although there were considerable differences—e.g., the number trimer discs in the lower core cylinders were different, and the relative positions of the some subunits, notably ApcD and ApcE, were different in the two. This would have to be addressed and corrected before this paper could be published.

Author reply 3

We apologize for the insufficient and incorrect citations in our original manuscript (particularly Zheng et al., *Nature Commun*, 2021.). We have confirmed the references suggested by reviewer 3 and cited them appropriately.

In the revised manuscript, we have discussed the fact that the number of APC trimers in the A (A') cylinder of the *T. vulcanus* PBS is different from other cyanobacteria PBSs (pages 6–7, lines 132–144). ApcD has been identified in the fourth of the A cylinders of the other cyanobacterial PBSs (Zheng et al., *Nature Commun*, 2021), and ApcD is contained in the *T. vulcanus* PBS (Supplementary Fig. 2C). We have performed a negative-staining EM 2D analysis to investigate the overall structure of more intact PBS, which keeps PC rods much better. The results show that the numbers of A-cylinders in the *T. vulcanus* PBS is three.

We validated the cryo-EM density map of all APC trimers and tentatively assigned the arrangement of ApcD. This site is the only one that could not be assigned as ApcA. It is difficult to assign the side chains of amino acid residues with the cryo-EM density map at a ~ 4.0 Å resolution, and this is also true for the map quality of Zheng et al. (see table below). The exact site of ApcD in PBS, irrespective of their source, needs to undergo experimental validation and will require a more detailed evaluation and discussion in the future.

	Validation by Q-score
ApcD (PDB code, Cryo-EM density map)	Estimated resolution (Å) (Q-score value)
T. vulcanus (7EVA, EMD-31944)	4.1 (0.39)

Nostoc 7120 (7EYD, EMD-31381)	4.0 (0.41)
Synechococcus 7002 (7EXT, EMD-31373)	4.3 (0.35)

Comment 4

The authors of this work seem to think for some reason that ApcF is a terminal emitter, although they provide no evidence for this. ApcF has been characterized first by Bryant et al. (1990) in Archives of Microbiology and by Biswas et al. (2010) in Applied and Environmental Microbiology. The protein has an absorbance maximum at 616 nm and fluorescence emission maximum at 637 nm, certainly not what one expects for a terminal emitter. It is only because of structural rearrangements in the core in apcF mutants that it seems to have effects on energy transfer.

Author reply 4

Stadnichuk et al. (Biochimica et Biophysica Acta, 1817, 1436–1445, 2012) and Jallet et al. (Biochimica et Biophysica Acta, 1817, 1418–1427, 2012) described ApcF as one of the candidates for a terminal emitter of PBS. However, as the reviewer commented, there is no direct experimental finding that ApcF is one of the terminal emitters. Hence, we decided to exclude ApcF as a terminal emitter.

Comment 5

Page 5. There is a discrepancy between the text a figure concerning the number of copies of ApcC. Are there six copies (text) or 4 copies (Figure 1G)?

Author reply 5

Thank you for indicating this error in Fig. 1G. The number of ApcCs identified in the *T. vulcanus* PBS is six. Fig. 1G has been now modified.

Comment 6

Page 5, lines 111-112. It is impossible to imagine that the structural differences would affect the efficiency of energy transfer significantly. If so, this difference would be selected by evolution and one structure would ultimately prevail. There may be detailed differences, but these are unlikely to affect the efficiency in any meaningful way.

Author reply 6

We agree with the reviewer's comment, and there are no experimental results that show that the efficiency of energy transfer depends on the numbers of APC trimers. We have therefore deleted this statement.

Comment 7

Page 6, lines 125 to 135. The relative positions of ApcD and ApcE in *T. vulcanus* and *Synechococcus* sp. PCC 7002 are different. Given the resolution, the assignments in this study are only tentative at best and possibly incorrect at worst. Not being able to identify a subunit as ApcA is not the same thing as positively identifying a subunit as ApcD. Authors should not assign any position as ApcD unless they can do so unequivocally, because it will only serve to introduce confusion into the field in the future. In other structures, the terminal emitters are assigned to adjacent trimers in the lower core cylinders with no separation. This must be addressed by the authors.

Author reply 7

As described in the above "Author reply 3," we have validated the cryo-EM density map of all APC trimers and tentatively assigned the arrangement of ApcD. This site is the only one that could not be assigned as ApcA. It is difficult to assign the side chains of amino acid residues with the cryo-EM density map at approximately a 4.0 Å resolution, and this is also true for the map quality of Zheng et al. (*Synechococcus* and *Nostoc*). Following the reviewer's comment, ApcD is shown in parentheses in all figures, and ApcA is assigned to the site in the structural model (PDB code: 7EVA).

Comment 8

Line 137. The authors must make clear that alpha-Lcm is a domain of ApcE (Lcm), just as the Rep domains are domains of the same protein.

Author reply 8

Accordingly, we have added the explanation for α^{LCM} (page 10, lines 171–172).

Comment 9

The authors provide no discussion for why their cores contain only 3 trimers, when nearly all other phycobilisomes studied have four trimers. No information about the purification is shown, and it is not possible to ascertain whether a trimer has been lost from the core during purification of the PBS. This seems rather likely, given that the peripheral rods are dissociating from the structures as well. If the authors are correct,

then a structural comparison to other cores with four trimers (as in the structures of Zheng et al.) can be undertaken to see what is different between the two cases. Because there are now two high-resolution structures (and was already a low-resolution structure) from different organisms with tricylindrical and pentacylindrical cores, the authors need to revise this paper completely to include comparisons to the other structure. That other paper was available to the authors prior to the submission of their paper, although probably not by much.

Author reply 9

According to this comment, we have added the following contents in the revised manuscript.

1) Sample preparation and dissociation of the PC rods from the intact PBS during preparation

In the prepared PBS, the PC rods interact with the PBS core. However, when using cryo-EM, it is difficult to obtain micrographs with a good contrast if the buffer solution contains stabilizing agents (in this case, the stabilizer was a phosphate buffer). Hence, it is necessary to remove the stabilizer before freezing the samples. This has been previously performed by Zheng et al. (*Synechococcus* PBS and *Nostoc* PBS), Ma et al. (*P. purpureum*), and Zhang et al. (*G. pacifica* PBS) in addition to this study. The interaction between the PC rods and PBS core in the PBSs of the two red algae was strong and remained as such even when the phosphate buffer concentration was reduced. However, the interaction between the PC rods and PBS core in the *T. vulcanus* PBS was weak, and the PC rods dissociated from the PBS core, although we replicated the grid preparation conditions presented by Zhang et al., 105, 57–63, 2017. We performed a 3D reconstruction of the dissociated PC rods obtained and analyzed them. (Supplementary Figs. 2 and 4; pages 5–6, lines 107–115).

We speculate that the differences in the structural stability of PBS (especially PC rods) for each species is attributable to the morphology of the PBSs. The hemidisoidal PBSs from cyanobacteria are easily dissociated. Therefore, we tried to stabilize the PBS structure using the GraFix approach that utilizes glutaraldehyde as a cross-linking agent. Zheng et al. also used glutaraldehyde to stabilize the structure of PBSs.

2) **The fourth APC trimer in the A cylinder is not present in the *T. vulcanus* PBS, unlike other cyanobacteria**

In other cyanobacterial PBSs, ApcD has been identified in the fourth APC trimer in the A cylinder. We thought that if the arrangement of ApcD is same in the PBSs of the cyanobacteria *Synechococcus*, *Nostoc*, and *T. vulcanus*, *T. vulcanus* PBS should have a fourth APC trimer in the A (A') cylinder.

We prepared more intact *T. vulcanus* PBS in a highly concentrated phosphate without GraFix treatment. Afterward, we observed this sample with negative staining EM and calculated 2D averages (Supplementary Fig. 2). The negative staining EM analysis reveals that the number of the APC trimers in the A cylinder of the *T. vulcanus* PBS is three. Moreover, the complexes of the PBS core and PC rods are likely maintained in these EM images, revealing that the samples are more intact. These results indicate that the number of the APC trimer in the A (A') cylinder of *T. vulcanus* PBS is three and not four, and ApcD likely resides in a different position from other cyanobacterial PBSs.

3) Structural comparison with the PBSs from other species

We have added the comparison sentences among the reported PBS structures to the revised manuscript (pages 18–20, lines 334–367). The overall structure of ApcE (L_{CM}), one of the important subunits in the PBS, is very similar although the species are different (Extended Table 5). Furthermore, ApcE among cyanobacteria has a high sequence identity. The C cylinder and Rep4 in L_{CM} , which stabilizes the structure, have only *T. vulcanus* and *Nostoc*, and the quality of the two structural models (7EVA and 7EYD) is low to enable a discussion of the detailed arrangement of the amino acid residues (~ 4 Å resolution; see Chain ID cA–cM and fA–fM in Supplementary Table 2). However, the small RMSD and high sequence identity between the two structures suggest that the functions of the C cylinder and Rep4 in the two species are similar.

In contrast, the energy transfer site to PSII in the L_{CM} was found to be both common and different in each species (Fig. 5). Tang et al. reported that the interaction of a Trp residue with the chromophore is a factor contributing to the red-shift in the absorption of the chromophore. This Trp is a conserved amino acid residue in all the species listed in the table, and it likely forms a π – π interaction with the chromophore in the αL_{CM} . However, this interaction has been detected only in the crystal structure (PDB code: 4XXI) and the *P. purpureum* PBS (PDB code:

6KGX). It is necessary to obtain high-resolution data for structural analysis to clarify the detailed interaction inside PBS.

While the importance of Trp interacting with the chromophore was the same in different species, the amino acid residue compositions around the chromophore varied slightly among species. The presence of amino acid residues with polar/charged groups affects the absorption energy of pigments (Saito et al., J. Photochem. Photobiol. A Chem., 402, 112799, 2020). Therefore, the amino acid residues around the chromophore may be a key factor in fine-tuning the energy level of the chromophore to ensure the unidirectional transfer of energy to PSII.

Comment 10

Considering the differences implied or confirmed, the authors need to account for why the pentacylindrical cores in *Anabaena/Nostoc* sp. PCC 7120 are different from those in *T. vulcanus*. There are four trimer equivalents in the core cylinders A/A' in 7120 but only 3 here. Why is that? How is that? Also, the positions of the ApcD subunits are different in the two structures. Why is that? If true, it would be the basis for an interesting comparison. If not the true, then the authors must explain why their results are different.

10. Page 10, lines 179-180. Again, the authors make an unjustified conclusion from the work of others. Methylation of the Asn residue in the beta subunits does not have a large effect on energy transfer. What it does instead is cause a sharp increase in ROS in cells, and that can be lethal in cells that don't tolerate ROS well (like *Synechocystis* sp. PCC 6803).

Author reply 10

As described in “Author reply 3,” the negative staining EM analysis of more intact PBS of *T. vulcanus* shows that the A (A') cylinder of the *T. vulcanus* PBS is three and not four, and ApcD likely resides in a different position from other cyanobacterial PBSs. At present, we do not clearly understand why *T. vulcanus* has three A (A') cylinders rather than four. As mentioned in the reviewer's comments, this is interesting and needs to be comprehensively investigated in the future.

Regarding the difference in the arrangement of ApcD, it is difficult to accurately identify the side chains of amino acid residues at approximately 4.0 Å resolution, and we consider that the arrangement of ApcD has not been accurately identified in the two cyanobacterial PBSs (see “Author reply 3”).

Regarding the sentence about the methylation of the Asn residue, we have checked the references on this again and found that the reviewer was right. Therefore, this sentence has been deleted.

Comment 11

11. Much of this paper is devoted to possible routes of excitation energy transfer. It is all just rank speculation, primarily based on distances, and completely ignoring any effects of strong coupling which there most certainly is in allophycocyanin. The energy transfer rates are sub-picosecond, and the absorption of AP and AP-B clearly shows the effects of strong excitonic coupling to the alpha subunit chromophores from the beta subunit of adjacent monomers in the trimers. This is not even considered in the discussion in this paper. The conclusion (Line 268) that ApcF is more important than ApcD and ApcE in terms of energy transfer defies decades of prior work showing otherwise.

Author reply 11

Thank you for the important remarks on the energy transfer in PBS. As the reviewer pointed out, the effect of exciton coupling is important for energy transfer in PBS.

Accordingly, we have revised the manuscript as follows:

However, the energy transfer between chromophores in the PBS cannot be explained by the FRET mechanism alone, because photoexcited coherence signals indicating intermolecular couplings are observed in both PC and APC^{44,45}. In addition, recent experiments revealed a new characteristic, in which excitations are coherently shared between donor and acceptor molecules during FRET⁴⁶. Moreover, previous studies suggest that excitonic coupling of some form may occur, even at distance of 20 Å to 30 Å, enabling the PBS to transfer energy at high efficiencies^{30,47}. MacColl⁴⁷ reported an excitonic coupling between the two chromophores of the APC trimer located in proximity across the monomer–monomer interface, and this strong interaction induces exciton to a red shift in the absorption spectrum. Although evidence supports both mechanisms, ultrafast fluorescence resulting from a number of approaches now indicate that the excitonic coupling mechanism is more plausible; however, although the structural model obtained in this study allows an interpretation of the arrangement and orientation of chromophores in the PBS and their surrounding interactions (especially with linker proteins), the structural model alone does not allow for a comprehensive interpretation of the exciton coupling between neighboring chromophores. The linker protein not only stabilizes the PBS structure but also alters the absorption/emission properties of the chromophore and may even create the necessary environment for exciton binding between chromophores³⁰. In this study, we propose the energy transfer pathway based on the distance between chromophores and

their orientation on the chromophores that interact with the linker proteins (pages 12–13, lines 221–239).

In addition, the description of ApcF has been revised as follows:

The protein environment around ${}^{\text{A}3}\beta_{\text{ApcF}}^{82}$ in ApcF appears to be one of the critical elements involved in transferring the unidirectional excitation energy to ${}^{\text{A}3}\alpha_{\text{LCM}}^{198}$ and eventually transferring the energy to PSII and PSI. (page 18, lines 325–327)

Comment 12

The discussion of PC rods is just an add-on thing because the rods were not stably attached to the cores. The authors don't say much about how the rods were obtained, how they were purified, and how the different CpcG paralogs affects the resolution. In the end, the discussion just ends abruptly with no concluding comments at all. There is more speculation about energy transfer, but the same issues that plague the discussion for cores are also problems with the peripheral rods. Chang et al. at least performed experiments to distinguish the CpcG paralogs functionally. No functional studies were described here.

Author reply 12

According to the reviewer comment, we have added the following details to our manuscript regarding the structural comparison of the PC rod [vs. crystal structure (David et al.; PDB: 3O2C) and the Nostoc PC rod (Zheng et al.)] (Fig. 7, Supplementary Fig. 15, Supplementary Table 7; pages 22–26, lines 399–458):

The PC rods dissociated from the PBS core contain multiple conformations (CpcG1, CpcG2, and CpcG4), and the overall structure of the linker protein is similar. Therefore, they could not be separated individually by the analysis of 3D classification. The details revealed in this study are different from the previously reported PC structure (symmetric structure) obtained by X-ray crystallography. The recently reported *Nostoc* PC rods form an asymmetric structure, as does the *T. vulcanus* PC rod. In particular, CpcD interacts with *T. vulcanus* PC rods, and the interaction of CpcD with PC rods changes the arrangement of PC monomer in the rod. In this study, we cannot propose functional differences between PC rods with different linker proteins; however, we have evaluated the interaction between PC monomers and linker proteins, albeit at a 4.2 Å resolution. This is a step forward from the work of David et al. (J. Mol. Biol., 405, 201–213, 2011) and Chang et al. (Cell Research, 25, 726–737, 2015).

Comment 13

Summary: Given that there are major differences among the structures of two red algal and two cyanobacterial phycobilisome structures and this one, the authors need to address these differences clear and discuss why these differences might exist. This is far more important than the speculative discussion concerning excitation energy transfer that basically reproduces similar published analyses. Why do the authors find only 3 trimers in the basal core cylinders, when other phycobilisomes have 4 trimers? That is a major question and having discrepancies like that in the literature is unacceptable unless they are explained. Otherwise, there will simply be confusion that is avoidable. It is possible now to directly compare the components/structures of two hemidisoidal phycobilisomes with pentacylindrical cores. How are those two structures similar and how do they differ? This is a far better basis for discussion than a speculative energy transfer scheme.

Author reply 13

We thank the reviewer for their insightful comments. According to the reviewer's comments, we have revised our manuscript accordingly and added a conclusion.

REVIEWER COMMENTS

Reviewer #1 (Remarks to the Author):

The resubmitted manuscript by Kawakami, Yonekura and co-workers, describing their determination of a cryo-EM structure, by single particle reconstruction of the core and rod of the Phycobilisome (PBS) antenna complex from the cyanobacterium *T. vulcanus* is greatly improved. They have corrected the omissions and mistakes in the cited literature, and also referred correctly to these papers to put their observations in perspective. I accept their explanation as to the reason for the additional lack of stability in the PBS structure, allowing the rods to easily disassociate. While temperature may also be in play (interactions may actually be strengthened at the 60^{>0}C growth temperature), this is only speculation on my part. The lack of existence of two forms of the PBS from *T. vulcanus*, is also acceptable, even though other claims (from the almost identical *T. elongatus* as well) exist.

One issue continues to trouble me. In the discussion of the rod structure, especially with respect to the comparison with the crystal structure of the rod, it is now shown that while the crystal structure is structurally homologous to the top hexamer (called by the authors disk A) of the cryo-EM rod structure, it is very different than that of the bottom hexamer (what is called by the authors disk b). The RMSD between the structures in this section are stated to be extremely large – up to >20Å! In no place is it stated by the authors on what level the comparison is made (α -carbons, backbone or whole atom). I hope that it is not an all atom comparison, as there is no sense in comparing such a low resolution cryo-EM structure with a high resolution crystal structure on this level. If it is α -carbons (which is what it should be), I must say that this is quite impossible. Such a difference would be found for two completely unrelated proteins. In Supplementary Table 7, the values of the superposition are given, showing not only is the crystal structure highly different than the present cryo-EM structure, but also that the present cryo-EM structure is similar to the 7EYD cryo-EM structure. I cannot see the present, unreleased EM structure in the PDB, but I can download the 7EYD, and from my measurements, there is no such difference between the crystal structure 3O2C and rod hexamers in 7EYD. In fact, if 3O2C is similar to the top hexamer, and different than the bottom hexamer (10 Å overall RMSD!), then there should be a similar difference between the two hexamers (top/bottom) in the cryo-EM structure. The authors are requested to check how they did the comparison. This issue must be investigated further prior to acceptance.

Suggested corrections.

On page 9, the authors mistakenly refer to the PDB deposition of the core structure as 7EVA and not the correct code – 7VEA.

Fig. 5F – The use of the capsule shaped objects (also used in Fig. 4) is confusing – and not explained in the legend. This is the most important panel in the figure. Perhaps just put in an arrow to show the direction.

Supp. Data Fig.10 legend – error in $\kappa^{>2}$.

Reviewer #2 (Remarks to the Author):

The revised manuscript entitled “Core and rod structures of a thermophilic cyanobacterial light-harvesting phycobilisome” by Kawakami et al. was improved in respect to the original version and addresses all my concerns, and, in my opinion, all the concerns of the other reviewers. The overall manuscript structure (especially the introduction, conclusion and methods) has been greatly improved and relevant literature is now correctly cited.

I just have one final comment:

- In Supp. Figure 2b, there seems to be something wrong with the top part of the gradient picture.

Reviewer #3 (Remarks to the Author):

See attached pdf file.

Core and rod structures of a thermophilic cyanobacterial light harvesting phycobilisome

Kawakami et al.

Comments on responses to reviewers 2 and 3

Page 8, author reply 9.

In the material added to the supplement, the authors now show negatively stained PBS with what appear to be a complete set of peripheral rods. Why weren't these PBS subjected to cryoEM analysis? Unless the rods dissociate from the cores during sample preparation, this would eliminate this necessity of solving rods separately from cores, wouldn't it?

Page 11-12, author reply 3.

Zheng et al. report their resolution as below 4.0 Å, namely 3.5 Å and 3.9 Å. They also solved the structure of a 7002 cpcC mutant. Considering that local resolution can be considerably better than the overall resolution, especially in the cores of structures, the actual resolution is considerably better than that described in the rebuttal, at least for global values in the core. The structure for those PBS cores agrees with prior detailed biochemical analyses as well. This allows Zheng et al. to identify the position of ApcD unambiguously. Thus, the position of ApcD, at least in three species with hemidisoidal PBS, is different from that found here. The authors have not provided any compelling answer to this question.

Page 12, author reply 4.

To my knowledge, Stadnichuk et al. had no special knowledge of the nature of ApcF as terminal emitter. They did not study the purified protein or complexes that contained it other than phycobilisomes. I previously provided information showing that this subunit was unlikely to be a terminal emitter. It is similar to other beta AP subunits. However, the chromophore on ApcF is excitonically coupled to the chromophore on ApcE, and as your Figure 5 shows, portions of ApcF are very close to the chromophore on ApcE. These two facts account for the impact of ApcF deletions on energy transfer via the chromophore of ApcE. Tang et al. completely missed the point that the chromophore of ApcE is excitonically coupled to that of ApcF because they crystallized and studied ApcE homodimers, not complexes in the trimeric state where ApcF can be in the proper location. In fact, no study has properly accounted for this important point. Because the chromophores of ApcE and ApcF are excitonically coupled, it is incorrect to describe energy transfer through one or the other of them. They act as a dimer for the purposes of energy transfer with two different absorbance bands of higher and lower energy. The excitonic coupling is an important aspect that is mostly missing when ApcF is replaced by ApcB in mutants when ApcF is deleted or inactivated. Studies conducted by Soulier and Bryant (2021) clearly demonstrated the effects of this excitonic coupling in a far-red light absorbing AP by mutating the chromophore binding cysteine residue in the partner allophycocyanin beta subunit, modifying the coupling.

Page 15, bottom of page.

As noted above, the structure of Tang et al. was determined for a homodimer, not a heterohexamer as would be required to understand more fully the chromophore environment of the PBP domain of ApcE. ApcF was not present in the studies of Tang et al., and yet Figure 5 shows that it makes important interactions in the area near the chromophore domain of ApcE. Thus, the nature of the

protein interactions that establish the properties of ApcE are better gained from the cryoEM structures than from the X-ray structures which sometimes can be very misleading. Soulier and Bryant (2021) have shown that the excitonic coupling of chromophores in allophycocyanins is a very important component of the factors that produce their specific spectroscopic features. Other authors have also studied this excitonic coupling. The key features are these excitonic properties, which delocalize the excitation over two chromophores, not one.

Page, 16, bottom of page.

In this reviewer's opinion, it is the position of ApcD in your study that is ambiguous and the positions in the two structures of Zheng et al. are defined. They additionally agree with detailed biochemical studies performed by Glazer and colleagues for cores from *Synechococcus* sp. PCC 6301/7942. This will be fully resolved by higher resolution structures, which are likely to agree with Zheng et al.

Page 18, top of page: As noted above, the chromophore on ApcF is excitonically coupled to the chromophore on ApcE. This coupling causes energy to be delocalized over both chromophores, so how is it then that one plays a more important role than the other? They act as a single entity with higher and lower energy absorbance bands delocalized over both phycocyanobilins. This is why the discussion of energy transfer pathways to the chromophore on ApcE make no sense.

Comments on the revised manuscript

The revised manuscript of Kawakami et al. is considerably improved over the previous version, and many of the comments by three reviewers, including those of this reviewer (#3), have been addressed in the revised manuscript.

In the opinion of this reviewer, the authors still have some problems, and most of the issues surround the inability to adequately describe why the core structure of this PBS is so different from that of *Anabaena/Nostoc* sp. PCC 7120.

Furthermore, the continued reliance on an overly simplistic view of energy transfer in the PBS is a problem that has not been addressed adequately in the revision.

Either there is loss of a trimer from the lower core cylinders, or there must be a difference in the ApcE polypeptide that accounts for a very significant rearrangement of the lower core cylinders. This remains an unanswered question, and the authors do not provide any compelling evidence that ApcE is sufficiently different to account for this difference.

There are also still numerous minor issues to be corrected (see below).

Lines 49-50. This sentence needs to be rewritten. Chlorophyll a has limited absorbance in the blue to orange region of the visible spectrum.

Line 50. This should be Phycobilisomes (plural), not singular. The authors are describing phycobilisomes generally as a class of objects, not just a single phycobilisome.

Lines 54-55. Oligomers (trimers, hexamers) of PBPs form without the necessity of linker proteins, which are used to join trimers and hexamers into larger scale structures (rods and cores).

Reference 15 is incorrect. It should be Glazer et al., 1979, Characterization of cyanobacterial phycobilisomes in zwitterionic detergents, *Proc. Natl. Acad. Sci. USA*, 76, 6162-6166 (1979).

Line 71. "Nostoc" is not a good shortened name for *Anabaena/Nostoc* sp. PCC 7120. A suitable name could be *Nostoc* 7120. This is because there are many *Nostoc* sp. and it should be made clearer which *Nostoc* sp. the authors are referring to. Similarly, *Synechococcus* sp. PCC 7002 can be referred to as

Synechococcus 7002, and Synechocystis sp. PCC 6803 can be referred to as Synechocystis 6803.

Line 82: should read: ...via thioether linkages to conserved cysteine residues...

Line 87: should read: ...which bind to the AP trimers of the core cylinders. Line 88-89: ...is rapidly transferred (on the order of 100 picoseconds)

Line 90: LCM should be renamed ApcE throughout the text. All the other subunits and proteins of the phycobilisome are named by the Demerec rules as the protein form of the gene name. There is no reason LCM should not be described in the same manner.

Line 110. Should read: PE rods, not PC rods Line 113: Replace "those" with "for the PE rods"

Line 123-124 is poorly structured. As it reads, it says that PC rods are part of the PBS core, which is clearly not the case.

Line 131 Replace 2 LCMs" with 2 ApcE (LCM),

Line 138: Synechococcus 7002? Nostoc 7120? Clarify.

Line 206-207. This statement requires a reference.

Line 250. Distance affects both rate and efficiency of energy transfer, not just efficiency.

Line 265: replace "composition" with "type"

Line 266: Should read: ... the number of the cysteine residue (84) to which... There aren't 84 residues to which chromophores are bound.

Lines 322 to 329. As noted in the comments above in response to the responses to reviewers, the chromophore on ApcF is excitonically coupled to the PCB chromophore on ApcE.

Figure 5 shows that ApcF interacts closely with the PCB on ApcE, and thus the excitation is delocalized over both chromophores. Deletion of apcF causes a major change in energy transfer and absorbance properties of ApcE for this reason, because ApcB differs from ApcF. This interaction with ApcF was missing in the X-ray structural study of Tang et al., and it is possible that Tang et al. drew incorrect conclusions (e.g., importance of pi-pi stacking) because of this difference.

Line 346: Synechococcus 7002; Nostoc 7120

Line 364: What does "Tyr79/ApcF was altered to Tyr" mean?

Lines 366-367. This sentence is confusing. Why should it affect the absorbance of the chromophore if it is too far away to form a pi-pi interaction? Magic?

Line 370: This statement is misleading. Each protomer is a heterodimer, which means that there would be 24 +3 subunits, not 12 +3. The rods have two dodecamers plus three linker polypeptides. If you want to describe protomers, be more specific.

Line 485 to 489. A structure cannot reveal energy transfer pathways. A structure and some analyses might suggest some alternative energy transfer pathways, but there is not a single pathway in a structure as complex as a phycobilisome. That already defies logic—and in particular defies the two-fold symmetry that demands at least TWO pathways at a minimum.

Line 497. "reasonably" is not a good word choice here, because it isn't reasonable for the authors to comment on what is or isn't reasonable. What would a significant or substantial difference be for the two structures? Has some objective criterion been met? Supplementary Table 6. There is a single entry for the chromophore on ApcD, and that is to an alpha-type subunit. This is not correct. The chromophore on ApcD is excitonically coupled to the chromophore on an ApcB subunit from an adjacent heterodimer (protomer) of AP. Excitation would be delocalized over both chromophores. In any Förster model, the alpha subunit chromophore, not the beta subunit chromophore, would be the acceptor, the opposite of the situation for phycocyanin. The donor alpha subunit chromophores act as donors to chlorophylls in PSI and PSII in a Förster model.

We would like to thank the reviewers for their insightful comments and kind and constructive suggestions to improve our manuscript. We have corrected the manuscript accordingly. All the reference papers suggested by the reviewers have been included in the revised manuscript.

Reviewer #1 (Remarks to the Author):

Comment 1

The resubmitted manuscript by Kawakami, Yonekura and co-workers, describing their determination of a cryo-EM structure, by single particle reconstruction of the core and rod of the Phycobilisome (PBS) antenna complex from the cyanobacterium *T. vulcanus* is greatly improved. They have corrected the omissions and mistakes in the cited literature, and also referred correctly to these papers to put their observations in perspective. I accept their explanation as to the reason for the additional lack of stability in the PBS structure, allowing the rods to easily disassociate. While temperature may also be in play (interactions may actually be strengthened at the 60°C growth temperature), this is only speculation on my part. The lack of existence of two forms of the PBS from *T. vulcanus*, is also acceptable, even though other claims (from the almost identical *T. elongatus* as well) exist.

Author's reply 1

Thank you for your positive comments. We agree with your suggestion that the growth conditions (temperature and light intensity) may alter the interaction between the PC rods in the PBS. We shall evaluate the effect of these conditions in the future studies.

Comment 2

One issue continues to trouble me. In the discussion of the rod structure, especially with respect to the comparison with the crystal structure of the rod, it is now shown that while the crystal structure is structurally homologous to the top hexamer (called by the authors disk A) of the cryo-EM rod structure, it is very different than that of the bottom hexamer (what is called by the authors disk b). The RMSD between the structures in this section are stated to be extremely large – up to >20Å!

Author's reply 2

Thank you for the important remark. In superposition of PC monomers in the PC rod, we superposed the C α backbone of the polypeptide in the PC monomer 1 (consisting of Chain A and Chain B) to the corresponding ones in *T. vulcanus* (3O2C) and *Nostoc* 7120 (7EYD). Then, RMSDs between the

other PC monomers were estimated, while maintaining the whole PC rod arrangement.

No significant difference in RMSDs was observed between PC monomers when superposing each PC monomer in the PC rod to the corresponding monomers in other species (probably the reviewer referred to this method for RMSD estimation). However, as in Supplementary Table 7, the whole the PC rod structures are different between the cryo-EM structure in this study and the X-ray structure that lacks the linker proteins (Ref. 33). Thus, the linker proteins in the PC rods change the arrangement of PC monomers and especially induce lateral shifts of PC monomers through their interaction between the PC monomers and the linker proteins. Based on the reviewer's comment, we have added the following description to our manuscript: We also added this explanation in Supplementary Table 7.

We superposed the $C\alpha$ backbone of the polypeptide in the PC monomer 1 (consisting of Chain A and Chain B) to the corresponding ones in *T. vulcanus* (3O2C) and *Nostoc* 7120 (7EYD). Then, RMSDs between the other PC monomers were estimated, while maintaining the whole PC rod arrangement. Although there were no major differences in the structure of each PC monomer comprising the PC rods, the interaction of the linker proteins in the PC rods caused a significant shift in the arrangement of Disk B in the PC rod. (page 24, lines 427-433)

Comment 3

In no place is it stated by the authors on what level the comparison is made (α -carbons, backbone or whole atom). I hope that it is not an all atom comparison, as there is no sense in comparing such a low resolution cryo-EM structure with a high resolution crystal structure on this level. If it is α -carbons (which is what it should be), I must say that this is quite impossible. Such a difference would be found for two completely unrelated proteins. In Supplementary Table 7, the values of the superposition are given, showing not only is the crystal structure highly different than the present cryo-EM structure, but also that the present cryo-EM structure is similar to the 7EYD cryo-EM structure. I cannot see the present, unreleased EM structure in the PDB, but I can download the 7EYD, and from my measurements, there is no such difference between the crystal structure 3O2C and rod hexamers in 7EYD. In fact, if 3O2C is similar to the top hexamer, and different than the bottom hexamer (10 Å overall RMSD!), then there should be a similar difference between the two hexamers (top/bottom) in the cryo-EM structure. The authors are requested to check how they did the comparison.

This issue must be investigated further prior to acceptance.

Author's reply 3

Thank you for your important remarks. Our response to this comment is mentioned in "Author's reply 2" and the following sentence have been added to our manuscript.

We superposed the C α backbone of the polypeptide in the PC monomer 1 (consisting of Chain A and Chain B) to the corresponding ones in *T. vulcanus* (3O2C) and *Nostoc* 7120 (7EYD). Then, RMSDs between the other PC monomers were estimated, while maintaining the whole PC rod arrangement. Although there were no major differences in the structure of each PC monomer comprising the PC rods, the interaction of the linker proteins in the PC rods caused a significant shift in the arrangement of Disk B in the PC rod. (page 24, lines 427-433)

Comment 4

On page 9, the authors mistakenly refer to the PDB deposition of the core structure as 7EVA and not the correct code – 7VEA.

Author's reply 4

We apologize for this oversight. The PDB code has been now corrected (7VEA) (page 9, line 166).

Comment 5

Fig. 5F – The use of the capsule shaped objects (also used in Fig. 4) is confusing – and not explained in the legend. This is the most important panel in the figure. Perhaps just put in an arrow to show the direction.

Author's reply 5

As for the display of the capsule-shaped objects (indicating that it is an alanine model) we would like to retain them in the figures. This display method was also used by Kawakami et al (PNAS, 2009. doi: 10.1073/pnas.0812797106.) and indicates the amino acid residues for which the side-chain arrangement cannot be precisely displayed. We have added a description of capsule-shaped object to the legends in Figs. 4–5 of our manuscript as below.

Amino acid residues for which the side-chain arrangement cannot be precisely displayed are indicated with capsule-shaped objects. The directions of atoms from C α to C β in the residues are indicated by the capsule-shaped objects. (page 18, lines 326-327; page 21, lines 371-374)

Comment 6

Supp. Data Fig.10 legend – error in κ^2 .

Author's to Reply 6

Thank you for highlighting the error. We have corrected this error in the revised manuscript.

Reviewer #2 (Remarks to the Author):

Comment 1

The revised manuscript entitled “Core and rod structures of a thermophilic cyanobacterial light-harvesting phycobilisome” by Kawakami et al. was improved in respect to the original version and addresses all my concerns, and, in my opinion, all the concerns of the other reviewers. The overall manuscript structure (especially the introduction, conclusion and methods) has been greatly improved and relevant literature is now correctly cited. I just have one final comment: - In Supp. Figure 2b, there seems to be something wrong with the top part of the gradient picture.

Author's reply 1

Thank you for your positive comments. We have now modified Supplementary Figure 2b based on your input.

Reviewer #3 (Remarks to the Author):

Comment 1

Page 8, author reply 9.

In the material added to the supplement, the authors now show negatively stained PBS with what appear to be a complete set of peripheral rods. Why weren't these PBS subjected to cryoEM analysis? Unless the rods dissociate from the cores during sample preparation, this would eliminate this necessity of solving rods separately from cores, wouldn't it?

Author's reply 1

First of all, we thank the reviewer for several important remarks regarding PBS. Uranyl acetate, one of the heavy atomic solutions, can quickly fix and stain samples, making it suitable for evaluating the purity and uniformity of unstable proteins such as PBS. However, the resolution limit of stain-treated samples is approximately 20 Å, at which, it is not possible to perform detailed 3D structural analysis of proteins, not only PBS.

Comment 2

Page 11-12, author reply 3.

Zheng et al. report their resolution as below 4.0 Å, namely 3.5 Å and 3.9 Å. They also solved the structure of a 7002 cpcC mutant. Considering that local resolution can be considerably better than the overall resolution, especially in the cores of structures, the actual resolution is considerably better than that described in the rebuttal, at least for global values in the core. The structure for those PBS cores agrees with prior detailed biochemical analyses as well. This allows Zheng et al. to identify the

position of ApcD unambiguously.

Author's reply 2

In cryo-EM analysis, local resolution is not necessarily higher than overall resolution (Gold standard Fourier Shell correlation criteria of 0.143 between two half maps). In protein structures analyzed by cryo-EM, regions of high local resolutions are generally located on the internal structure, while the local resolution of the outer protein regions tends to be lower than the overall resolution. This could be attributed to the fact that the outer regions are in contact with the solvent and are therefore, more likely to be disordered. As indicated in our first response, we evaluated the agreement between ApcD (*Nostoc* 7120 and *Synechococcus* 7002) and its maps identified by Zheng et al. et al. using Q-score, and the values were 0.41 (estimated resolution = 4.0 Å) and 0.35 (estimated resolution = 4.3 Å), respectively. Q-score is a quantitative parameter to characterize the resolvability of individual atoms in cryo-EM maps (Pintilie et al. Nat. Methods., 2020). The Q-score clearly indicated that the resolution of the two maps identified as ApcD were lower than the overall resolutions (3.5 Å and 3.9 Å).

This explanation does not mean that ApcD identification by Zheng et al. was wrong. However, as indicated in the first response, it is difficult to identify the side-chains of amino acid residues at approximately 4.0 Å resolution. Hence, Zheng et al. were not able to locate ApcD by cryo-EM analysis “unambiguously”.

Identification of protein subunit arrangement via biochemical analysis is challenging. Earlier, Tal et al. (JBC, 2014. doi: 10.1074/jbc.M114.595942.) analyzed the subunit arrangement by cross-linking *T. vulcanus* PBS and reported that the Lys residues of ApcD crosslinked with Arg433 (in Rep1), Arg490 (in Rep2), and Arg729 (in Rep3) of ApcE. However, the ApcD arrangement proposed by them is inconsistent not only with that proposed in this study, but also with the one proposed by Zhang et al. Thus, to accurately identify subunit arrangement in a protein, it is critical to evaluate the side-chains of individual amino acid residues from the 3D map. Unfortunately, this can be challenging with a 3D map at a resolution lower than 3 Å.

Comment 3

Thus, the position of ApcD, at least in three species with hemidiscoidal PBS, is different from that found here. The authors have not provided any compelling answer to this question.

Author's reply 3

According to the reviewer's comment, we have added the following description in the revised manuscript:

The lack of the "fourth APC trimer" in the A cylinder of *T. vulcanus* PBS may be attributed to the fact that ApcC interacting in the APC trimer is unable to interact with Rep1 in ApcE. There are two main conformation types (Type I and II) in ApcC of the PBS present in *Nostoc* 7120 and *Synechococcus* 7002. ApcC that interacts with Reps2–4 is type I, whereas the ApcC that interacts with Rep1 is Type II (Supplementary Fig. 10). The structures of ApcC (Types I and II) of *Nostoc* 7120 and *Synechococcus* 7002 are similar (RMSD: 0.78 [Type I] and 0.74 [Type II]). In contrast, in *T. vulcanus* PBS, ApcC interacting with Reps3–4 is Type I, but ApcC interacting with Rep2 is neither Type I nor II (that is, Type III). Comparison of Types II and III revealed different arrangement of the loop regions in ApcC (residues 9–26, Supplementary Fig. 10c), and slightly larger RMSD between Types I and III (1.7). Although the amino acid sequences of Repls1–4 and ApcC are similar in the three cyanobacteria (Supplementary Fig. 10d), the environment around Rep2 in *T. vulcanus* PBS may alter the interaction of ApcC compared with that in the other cyanobacteria. However, it is also possible that the environment surrounding Rep1 in *T. vulcanus* PBS does not allow ApcC interaction, and hence, the "fourth APC trimer" is unable to interact with Rep1. (page 12 lines 210-224)

The amino acid sequence of ApcE (Reps 1–4) is similar in the three cyanobacteria, as is the arrangement of C α in the structural models (see Supplementary Table 5). In addition, the amino acid sequence of ApcC is similar in the three cyanobacteria (see Supplementary Fig. 10d). However, *T. vulcanus* PBS core contains a ApcC type (Type III) not found in the two cyanobacterial species. Considering the altered position of C α within ApcC that interacts with Rep2, it is possible that the environment surrounding Rep1 prevents the formation of interactions between Rep1 and ApcC.

Comment 4

Page 12, author reply 4.

To my knowledge, Stadnichuk et al. had no special knowledge of the nature of ApcF as terminal emitter. They did not study the purified protein or complexes that contained it other than phycobilisomes. I previously provided information showing that this subunit was unlikely to be a terminal emitter. It is similar to other beta AP subunits. However, the chromophore on ApcF is excitonically coupled to the chromophore on ApcE, and as your Figure 5 shows, portions of ApcF are very close to the chromophore on ApcE. These two facts account for the impact of ApcF deletions on energy transfer via the chromophore of ApcE. Tang et al. completely missed the point that the chromophore of ApcE is excitonically coupled to that of ApcF because they crystallized and studied ApcE homodimers, not complexes in the trimeric state where ApcF can be in the proper location. In fact, no study has properly accounted for this important point. Because the chromophores of ApcE and ApcF are excitonically coupled, it is incorrect to describe energy transfer through one or the other of them. They act as a dimer for the purposes of energy transfer with two different absorbance bands of higher and lower energy. The excitonic coupling is an important aspect that is mostly missing when ApcF is replaced by ApcB in mutants when *apcF* is deleted or inactivated. Studies conducted by Soulier and Bryant (2021) clearly demonstrated the effects of this excitonic coupling in a far-red light absorbing AP by mutating the chromophore binding cysteine

residue in the partner allophycocyanin beta subunit, modifying the coupling.

Author's reply 4

Thank you for the important remarks on excitonic coupling between the chromophores in ApcE and ApcF. Accordingly, we have modified the following sentences in the 2nd draft of the revised manuscript.

The environment around ${}^{A3}\beta_{\text{ApcF}}^{82}$ in ApcF and ${}^{A3}\alpha_{\text{LCM}}^{198}$ in ApcE may be one of the critical elements involved in transferring the unidirectional excitation energy to PSI and PSII. Soulier and Bryant reported that the interactions between PCBs bound to α and β subunits of adjacent APC monomer in APC trimer are responsible for the red-shift in the chromophore absorption⁵¹. In the *T. vulcanus* PBS, the distance between ${}^{A3}\beta_{\text{ApcF}}^{82}$ and ${}^{A3}\alpha_{\text{LCM}}^{198}$ is 20 Å, suggesting that the two chromophores could form an excitonic coupling that eventually transfers excitation energy to PSI and PSII. ApcD is a terminal emitter that transfers energy to PSI⁴⁸⁻⁵⁰, and the findings of the present study suggest that ${}^{A1}\alpha_{\text{ApcD}}^{81}$ forms an excitonic coupling with the neighboring ${}^{A1}\beta_1^{84}$ and transfers excitation energy to PSI. (pages 19 lines 343-351)

Comment 5

Page 15, bottom of page.

As noted above, the structure of Tang et al. was determined for a homodimer, not a heterohexamer as would be required to understand more fully the chromophore environment of the PBP domain of ApcE. ApcF was not present in the studies of Tang et al., and yet Figure 5 shows that it makes important interactions in the area near the chromophore domain of ApcE. Thus, the nature of the protein interactions that establish the properties of ApcE are better gained from the cryoEM structures than from the X-ray structures which sometimes can be very misleading. Soulier and Bryant (2021) have shown that the excitonic coupling of chromophores in allophycocyanins is a very important component of the factors that produce their specific spectroscopic features. Other authors have also studied this excitonic coupling. The key features are these excitonic properties, which delocalize the excitation over two chromophores, not one.

Author's reply 5

As described in "Author's reply 4", the sentence on excitonic coupling interaction of the chromophores (${}^{A3}\beta_{\text{ApcF}}^{82}$ and ${}^{A3}\alpha_{\text{LCM}}^{198}$) has been described in our manuscript (pages 19 lines 343-351). Tang et al. indeed reported a "homodimeric α -domain structure", and not the native structure of the PBS core (interaction with ApcE and ApcF). However, since Trp, which forms a π - π interaction with ${}^{A3}\alpha_{\text{LCM}}^{198}$, is conserved in all species, it is suggested that this interaction causes the red-shift of the chromophore. Hence, we have retained the description of the Tang et al. study in our manuscript.

Comment 6

Page, 16, bottom of page.

In this reviewer's opinion, it is the position of ApcD in your study that is ambiguous and the positions in the two structures of Zheng et al. are defined. They additionally agree with detailed biochemical studies performed by Glazer and colleagues for cores from *Synechococcus* sp. PCC 6301/7942. This will be fully resolved by higher resolution structures, which are likely to agree with Zheng et al.

Author's reply 6

We evaluated the cryo-EM map of the A (A') cylinder in detail and identified the "non-ApcA subunit" (Supplementary Fig. 6). Reviewer 2 has kindly agreed to this approach. However, as Reviewer 3 pointed out, we did not "unambiguously identify ApcD," therefore, we have indicated ApcD in parentheses in the figures. In addition, we have shown that the A (A') cylinder of *T. vulcanus* PBS (a more intact structure with interacting PC rods) consists of three APC trimers even after sample preparation (Supplementary Figure 2).

Again, we do not suggest that Zheng et al.'s identification of ApcD is incorrect. To begin with, *Nostoc* 7120 (or *Synechococcus* 7002) and *T. vulcanus* are different species. However, as indicated in the first response and "Author's reply 2", it is difficult to identify the side-chains of amino acid residues at a resolution of approximately 4.0 Å.

Comment 7

Page 18, top of page: As noted above, the chromophore on ApcF is excitonically coupled to the chromophore on ApcE. This coupling causes energy to be delocalized over both chromophores, so how is it then that one plays a more important role than the other? They act as a single entity with higher and lower energy absorbance bands delocalized over both phycocyanobilins. This is why the discussion of energy transfer pathways to the chromophore on ApcE make no sense.

Furthermore, the continued reliance on an overly simplistic view of energy transfer in the PBS is a problem that has not been addressed adequately in the revision.

Author's reply 7

Thank you for the important remarks on excitonic coupling between the chromophores in ApcE and ApcF. As mentioned in "Author's Reply 4", we have modified the following sentences in the 2nd revised manuscript.

The environment around $A^3\beta_{ApcF}^{82}$ in ApcF and $A^3\alpha_{LCM}^{198}$ in ApcE may be one of the critical elements involved in transferring the unidirectional excitation energy to PSI and PSII. Soulier and Bryant reported that the interactions between PCBs bound to α and β subunits of adjacent APC

monomer in APC trimer are responsible for the red-shift in the chromophore absorption⁵¹. In the *T. vulcanus* PBS, the distance between $^A\beta_{\text{ApcF}}^{82}$ and $^A\alpha_{\text{LCM}}^{198}$ is 20 Å, suggesting that the two chromophores could form an excitonic coupling that eventually transfers excitation energy to PSI and PSII. ApcD is a terminal emitter that transfers energy to PSI⁴⁸⁻⁵⁰, and the findings of the present study suggest that $^A\alpha_{\text{ApcD}}^{81}$ forms an excitonic coupling with the neighboring $^A\beta_1^{84}$ and transfers excitation energy to PSI. (pages 19 lines 343-351)

Comment 8

Either there is loss of a trimer from the lower core cylinders, or there must be a difference in the ApcE polypeptide that accounts for a very significant rearrangement of the lower core cylinders. This remains an unanswered question, and the authors do not provide any compelling evidence that ApcE is sufficiently different to account for this difference.

Author's reply 8

Thank you for the important remarks. The following sentences have been included in our manuscript to explain the interaction between ApcE (Reps1-4) and ApcC.

The lack of the "fourth APC trimer" in the A cylinder of *T. vulcanus* PBS may be attributed to the fact that ApcC interacting in the APC trimer is unable to interact with Rep1 in ApcE. There are two main conformation types (Type I and II) in ApcC of the PBS present in *Nostoc* 7120 and *Synechococcus* 7002. ApcC that interacts with Reps2-4 is type I, whereas the ApcC that interacts with Rep1 is Type II (Supplementary Fig. 10). The structures of ApcC (Types I and II) of *Nostoc* 7120 and *Synechococcus* 7002 are similar (RMSD: 0.78 [Type I] and 0.74 [Type II]). In contrast, in *T. vulcanus* PBS, ApcC interacting with Reps3-4 is Type I, but ApcC interacting with Rep2 is neither Type I nor II (that is, Type III). Comparison of Types II and III revealed different arrangement of the loop regions in ApcC (residues 9-26, Supplementary Fig. 10c), and slightly larger RMSD between Types I and III (1.7). Although the amino acid sequences of Reps1-4 and ApcC are similar in the three cyanobacteria (Supplementary Fig. 10d), the environment around Rep2 in *T. vulcanus* PBS may alter the interaction of ApcC compared with that in the other cyanobacteria. However, it is also possible that the environment surrounding Rep1 in *T. vulcanus* PBS does not allow ApcC interaction, and hence, the "fourth APC trimer" is unable to interact with Rep1. (page 12, lines 210-224)

Comment 9 and 10

Lines 49-50. This sentence needs to be rewritten. Chlorophyll a has limited absorbance in the blue to orange region of the visible spectrum.

Line 50. This should be Phycobilisomes (plural), not singular. The authors are describing phycobilisomes generally as a class of objects, not just a single phycobilisome.

Author's replies 9 and 10

According to the Reviewer's comment, we have modified this sentence as follows:

PBSs absorb mainly visible light at 490-650 nm, which PSI and PSII have difficulty absorbing (as an exception, cyanobacteria containing chlorophyll *f* harbor PBSs that absorb near-infrared light^{2,3})

(page 3, lines 48-50)

Comment 11

Lines 54-55. Oligomers (trimers, hexamers) of PBPs form without the necessity of linker proteins, which are used to join trimers and hexamers into larger scale structures (rods and cores).

Author's reply 11

According to the Reviewer's comment, we have modified this sentence as follows:

Oligomers of these α - and β -subunits interact with non-chromophorylated linker proteins to form PC rods in the whole PBS complex. (page 3, lines 55-56)

Comment 12

Reference 15 is incorrect. It should be Glazer et al., 1979, Characterization of cyanobacterial phycobilisomes in zwitterionic detergents, Proc. Natl. Acad. Sci. USA, 76, 6162-6166 (1979).

Author's reply 12

We apologize for this oversight. We have corrected the citation (Ref. 15). (page 39, lines 744-746)

Comment 13

Line 71. "Nostoc" is not a good shortened name for *Anabaena/Nostoc* sp. PCC 7120. A suitable name could be *Nostoc* 7120. This is because there are many *Nostoc* sp. and it should be made clearer which *Nostoc* sp. the authors are referring to. Similarly, *Synechococcus* sp. PCC 7002 can be referred to as *Synechococcus* 7002, and *Synechocystis* sp. PCC 6803 can be referred to as *Synechocystis* 6803.

Author's reply 13

We have modified the shortened names per your suggestion in our manuscript (*Nostoc* 7120 and *Synechococcus* 7002).

Comments 14-19

Line 82: should read: ...via thioether linkages to conserved cysteine residues...

Line 87: should read: ...which bind to the AP trimers of the core cylinders. Line 88-89: ...is rapidly transferred (on the order of 100 picoseconds)

Line 90: LCM should be renamed ApcE throughout the text. All the other subunits and proteins of the phycobilisome are named by the Demerec rules as the protein form of the gene name. There is no reason LCM should not be described in the same manner.

Line 110. Should read: PE rods, not PC rods

Line 113: Replace “those” with “for the PE rods”

Line 131 Replace 2 LCMs” with 2 ApcE (LCM),

Author’s replies 14–19

Thank you for suggesting these revisions. We have revised the manuscript accordingly.

Comment 20

Line 123-124 is poorly structured. As it reads, it says that PC rods are part of the PBS core, which is clearly not the case.

Author’s reply 20

We have modified the sentence as follows:

The analyzed PBS core of *T. vulcanus* shows a hemidiscoidal structure with C2 symmetry, composed of three cylinders (A, A’, and B), two cylinders (C and C’) with some PC rods interacting with the PBS core (Fig. 1). (page 6 lines 123-125).

Comment 21

Line 138: Synechococcus 7002? Nostoc 7120? Clarify.

Author’s reply 21

We have modified the shortened names as *Nostoc* sp. PCC7120 and *Synechococcus* sp. PCC 7002.

Comment 22

Line 206-207. This statement requires a reference.

Author’s reply 22

We have added the citation (Ref. 1) to support the indicated statement. (page 12, line 207)

Reference 1: Sidler, W. A. *Phycobilisome and Phycobiliprotein Structures* (Advanced Photosynthesis and Respiration, Volume 1, Springer, Dordrecht), pp. 139–216.

Comment 23

Line 250. Distance affects both rate and efficiency of energy transfer, not just efficiency.

Author’s reply 23

Thank you for highlighting this deficiency. We have added “rate” to this sentence. (page 14 lines

265)

Comments 24 and 25

Line 265: replace “composition” with “type”

Line 266: Should read: ... the number of the cysteine residue (84) to which... There aren't 84 residues to which chromophores are bound.

Author's replies 24 and 25

We have modified the sentence as follows:

... subunit type (α or β), and the number of cysteine residue (84) to which ... (pages 16 lines 281-282)

Comment 26

Lines 322 to 329. As noted in the comments above in response to the responses to reviewers, the chromophore on ApcF is excitonically coupled to the PCB chromophore on ApcE.

Author's reply 26

In the revised draft, we have modified the sentences as follows:

The environment around ${}^{\text{A}3}\beta_{\text{ApcF}}{}^{82}$ in ApcF and ${}^{\text{A}3}\alpha_{\text{LCM}}{}^{198}$ in ApcE may be one of the critical elements involved in transferring the unidirectional excitation energy to PSI and PSII. Soulier and Bryant reported that the interactions between PCBs bound to α and β subunits of adjacent APC monomer in APC trimer are responsible for the red-shift in the chromophore absorption⁵¹. In the *T. vulcanus* PBS, the distance between ${}^{\text{A}3}\beta_{\text{ApcF}}{}^{82}$ and ${}^{\text{A}3}\alpha_{\text{LCM}}{}^{198}$ is 20 Å, suggesting that the two chromophores could form an excitonic coupling that eventually transfers excitation energy to PSI and PSII. ApcD is a terminal emitter that transfers energy to PSI⁴⁸⁻⁵⁰, and the findings of the present study suggest that ${}^{\text{A}1}\alpha_{\text{ApcD}}{}^{81}$ forms an excitonic coupling with the neighboring ${}^{\text{A}1}\beta_1{}^{84}$ and transfers excitation energy to PSI. (page 19, lines 343-351)

Comment 27

Figure 5 shows that ApcF interacts closely with the PCB on ApcE, and thus the excitation is delocalized over both chromophores. Deletion of apcF causes a major change in energy transfer and absorbance properties of ApcE for this reason, because ApcB differs from ApcF. This interaction with ApcF was missing in the X-ray structural study of Tang et al., and it is possible that Tang et al. drew incorrect conclusions (e.g., importance of pi-pi stacking) because of this difference.

Author's reply 27

Tang et al reported a "homodimeric α -domain structure", and not the native structure in the PBS core (interaction with ApcE and ApcF). However, since Trp, which forms a π - π interaction with ${}^{\text{A}3}\alpha_{\text{LCM}}{}^{198}$, is conserved in all species, and it is suggest that this interaction causes the red-sift of the chromophore, we have retained the description of the Tang et al. study in the revised manuscript.

Comment 28

Line 346: *Synechococcus* 7002; *Nostoc* 7120

Author's reply 28

We have modified the shortened names in our manuscript (*Nostoc* 7120 and *Synechococcus* 7002).

Comment 29

Line 364: What does "Tyr79/ApcF was altered to Tyr" mean?

Author's reply 29

In the revised manuscript, this sentence has been modified as follows:

the amino acid residue corresponding to Tyr79/ApcF was Tyr (*G. pacifica* and *P. purpureum*), Leu (*Nostoc* 7120), and Phe (*Synechococcus* 7002) in each species. (pages 22 lines 389-390)

Comment 30

Lines 366-367. This sentence is confusing. Why should it affect the absorbance of the chromophore if it is too far away to form a pi-pi interaction? Magic?

Author's reply 30

We apologize for the confusing description. Zheng et al proposed that "The residues Phe79 and Phe60 (in *Synechococcus* 7002) are located between the bilin of ApcF (β_{ApcF}) and α_{ApcE} on the trimers A3/A'3. Their distances to the α_{ApcE} are both within 5.0 Å, capable of forming π - π interaction with the bilin."

However, as can be observed in Fig. 5, Phe79 and Phe60 cannot form π - π interactions with the chromophore. However, as Saito et al. (Ref. 41) have reported, polar/charged amino acid residues affect the absorption energy of pigments, we have modified this sentence as follows:

This residue is located a distance from the chromophore ring that makes it unlikely to form a π - π interaction; however, it may affect the absorption property of the chromophore because polar/charged amino acid residues affect the absorption energy of pigments⁴¹. (pages 22 lines 390-393)

Comment 31

Line 370: This statement is misleading. Each protomer is a heterodimer, which means that there would be 24 +3 subunits, not 12 +3. The rods have two dodecamers plus three linker polypeptides. If you want to describe protomers, be more specific.

Author's reply 31

We have modified the sentence as follows:

The PC rod are formed by PC monomers consisting of α -subunit (ApcA) and β -subunit (ApcB) to form two PC hexamers, inside which linker proteins interact. (page 22, lines 396-397)

Comment 32

Line 485 to 489. A structure cannot reveal energy transfer pathways. A structure and some analyses might suggest some alternative energy transfer pathways, but there is not a single pathway in a structure as complex as a phycobilisome. That already defies logic—and in particular defies the two-fold symmetry that demands at least TWO pathways at a minimum.

Author's reply 32

We have modified the sentence as follows:

In summary, cryo-EM analysis of the PBS core and PC rod from *T. vulcanus* (a hemidiscoidal structure in many cyanobacterial species) revealed the chromophore arrangement in the PBS and its surrounding structure.. (pages 29 lines 517-519)

Comment 33

Line 497. “reasonably” is not a good word choice here, because it isn't reasonable for the authors to comment on what is or isn't reasonable. What would a significant or substantial difference be for the two structures? Has some objective criterion been met?

Author's reply 33

We agree with the reviewer's comment. We have deleted this word “reasonably.”

Comment 34

Supplementary Table 6. There is a single entry for the chromophore on ApcD, and that is to an alpha-type subunit. This is not correct. The chromophore on ApcD is excitonically coupled to the chromophore on an ApcB subunit from an adjacent heterodimer (protomer) of AP. Excitation would be delocalized over both chromophores. In any Förster model, the alpha subunit chromophore, not the beta subunit chromophore, would be the acceptor, the opposite of the situation for phycocyanin. The donor alpha subunit chromophores act as donors to chlorophylls in PSI and PSII in a Förster model.

Author's reply 34

Thank you for the important remarks. Accordingly, we have modified Supplementary Table 6 in line with your suggestion.

REVIEWERS' COMMENTS

Reviewer #1 (Remarks to the Author):

The authors have answered my queries to the previous version successfully. My only comment now is that is Fig. 7A (and legend) which relates to the structure of the rods - the linker is denoted as ApcE - and not CpcD which is the correct rod capping linker.

Reviewer #3 (Remarks to the Author):

The authors have done a thorough job in responding to the three reviewers, and the changes made in the manuscript are acceptable and have improved the presentation considerably.

I have only two remaining comments.

Firstly, concerning the response to comment 1 to Reviewer 3: I was not suggesting that the authors determine the structure from negatively stained material. Uranyl acetate is not a fixative, but there is no reason why the intact PBS shown could not be subjected to cryoEM analysis.

Secondly, concerning the response to comment 31 and the text now in lines 396 and 397 of the manuscript: the genes encoding the alpha and beta subunits of phycocyanin are *cpcA* and *cpcB*, not *apcA* and *apcB*. The corresponding proteins should be CpcA and CpcB. This should be corrected.

We thank the reviewers for their kind comments to improve our manuscript. We have corrected the manuscript accordingly.

Reviewer #1 (Remarks to the Author):

Comment 1

The authors have answered my queries to the previous version successfully. My only comment now is that in Fig. 7A (and legend) which relates to the structure of the rods - the linker is denoted as ApcE - and not CpcD which is the correct rod capping linker.

Author's reply 1

Thank you for indicating the error. We have corrected this error in the revised manuscript.

Reviewer #3 (Remarks to the Author):

Comment 1

Firstly, concerning the response to comment 1 to Reviewer 3: I was not suggesting that the authors determine the structure from negatively stained material. Uranyl acetate is not a fixative, but there is no reason why the intact PBS shown could not be subjected to cryoEM analysis.

Author's reply 1

We apologize for misunderstanding the reviewer's comment. As mentioned in the first response, for cryo-EM imaging, it is preferable to remove highly concentrated stabilizers (in this case, the stabilizer was a phosphate buffer), but this promotes the dissociation of PC rods from the PBS core. Hence, it is necessary to fix the sample for structural analysis of PBS using cryo-EM. It is possible to collect images of uranyl-stained PBS using cryo-EM. However, as mentioned in the second response, the resolution of these stained samples is limited (typically to 20 Å). Therefore, it is impossible to discuss detailed subunit arrangements and interactions from the stained PBS.

Comment 2

Secondly, concerning the response to comment 31 and the text now in lines 396 and 397 of the manuscript: the genes encoding the alpha and beta subunits of phycocyanin are *cpcA* and *cpcB*, not *apcA* and *apcB*. The corresponding proteins should be CpcA and

CpcB. This should be corrected.

Author's reply 2

Thank you for indicating the error. We have corrected this error in the revised manuscript.